# Slow Transition to Low-Dimensional Chaos in Heavy-Tailed Recurrent Neural Networks

**Eva Yi Xie[1,2]    Stefan Mihalas[1]    Łukasz Kuśmierz[1]**
[1] Allen Institute, Seattle, WA, USA
[2] Princeton Neuroscience Institute, Princeton University, NJ, USA
evayixie@princeton.edu, {stefanm, lukasz.kusmierz}@alleninstitute.org

## Abstract

Growing evidence suggests that synaptic weights in the brain follow heavy-tailed distributions, yet most theoretical analyses of recurrent neural networks (RNNs) assume Gaussian connectivity. We systematically study the activity of RNNs with random weights drawn from biologically plausible Lévy alpha-stable distributions. While mean-field theory for the infinite system predicts that the quiescent state is always unstable—implying ubiquitous chaos—our finite-size analysis reveals a sharp transition between quiescent and chaotic dynamics. We theoretically predict the gain at which the finite system transitions from quiescent to chaotic dynamics, and validate it through simulations. Compared to Gaussian networks, finite heavy-tailed RNNs exhibit a broader gain regime near the edge of chaos, namely, a slow transition to chaos. However, this robustness comes with a tradeoff: heavier tails reduce the Lyapunov dimension of the attractor, indicating lower effective dimensionality. Our results reveal a biologically aligned tradeoff between the robustness of dynamics near the edge of chaos and the richness of high-dimensional neural activity. By analytically characterizing the transition point in finite-size networks—where mean-field theory breaks down—we provide a tractable framework for understanding dynamics in realistically sized, heavy-tailed neural circuits.[*]

## 1   Introduction

Advances in connectomics yield increasingly detailed wiring diagrams of neural circuits across species and brain regions [Dorkenwald et al., 2024, The MICrONS Consortium, 2025]. This progress raises fundamental questions: what structural principles govern neural circuits, and how do they support the brain's remarkable computational power? One such prominent structural feature is the presence of heavy-tailed [Foss et al., 2011] synaptic weight distributions, consistently observed across the mammalian cortex [Song et al., 2005, Lefort et al., 2009, Dorkenwald et al., 2022], mammalian hippocampus [Ikegaya et al., 2013], and even in the *Drosophila* central brain [Scheffer et al., 2020]. Notably, this feature stands in sharp contrast to the Gaussian weight assumptions that dominate theoretical neuroscience studies and light-tailed distributions utilized in the standard initialization schemes in modern artificial neural networks [LeCun et al., 2002, Glorot and Bengio, 2010, He et al., 2015]. One way to formally model heavy tails is with the family of Lévy $\alpha$-stable distributions [Feller, 1971, Borak et al., 2005], which emerges as a natural generalization of the familiar Gaussian distribution ($\alpha = 2$) via the generalized central limit theorem. The family is parameterized by a stability index $\alpha$, where smaller values of $\alpha$ correspond to heavier tails. For $\alpha < 2$, these distributions feature heavy, power-law tails. Similarly to experimentally measured synaptic weights, samples generated from such distributions consistently contain large outliers that dominate many sample statistics. As we show in this paper, this can strongly affect neural dynamics.

---

[*]The codebase is publicly available at `https://github.com/AllenInstitute/HeavyRNN_public`.

39th Conference on Neural Information Processing Systems (NeurIPS 2025).

One key phenomenon studied in theoretical neuroscience and machine learning is the transition to chaos, which has long been hypothesized to support optimal information flow and computational capacity at the so-called *edge of chaos* in a wide variety of randomly initialized neural networks. This encompasses recurrent neural networks (RNNs) [Bertschinger et al., 2004, Legenstein and Maass, 2007, Toyoizumi and Abbott, 2011, Schuecker et al., 2018] and feedforward neural networks [Schoenholz et al., 2016, Poole et al., 2016]. In particular, Schoenholz et al. [2016] shows that the trainability of deep networks depends on initializing near the edge of chaos: the farther from this critical regime, the shallower a network must be to remain trainable. Similarly, Bertschinger et al. [2004] shows that only near the edge of chaos can RNNs perform complex computations on time series.

In the context of feedforward networks, this effect can be understood in terms of the neural network Gaussian process kernel [Neal, 1996]: outside of the critical point, analogous to the edge of chaos in RNNs, a trivial fixed point with kernel constant almost everywhere is approached exponentially fast, limiting the effective depth of information propagation and network trainability [Schoenholz et al., 2016, Lee et al., 2017]. The transition between chaotic and non-chaotic dynamics in RNNs is often discussed in terms of eigenvalues of the weight matrix [Rajan and Abbott, 2006, Aljadeff et al., 2015]. According to the circular law [Girko, 1985, Tao et al., 2010], its eigenvalues are bounded in a circle of radius proportional to the standard deviation of the distribution of weight entries. If the standard deviation is large enough, some eigenvalues fall outside of the unit circle and the quiescent state becomes unstable, paving the way for chaos to emerge. In contrast to random matrices with light-tailed elements, random matrices with $\alpha$-stable entries feature an unbounded limiting density of eigenvalues [Bordenave et al., 2011]. In infinite networks, this can lead to the lack of transition between quiescent and chaotic states, with any perturbation ultimately expanding in a chaotic manner [Kuśmierz et al., 2020].

Understanding the computational implications of heavy-tailed recurrent connectivity is especially timely as RNNs have become central to neuroscience modeling. They are used to reproduce latent trajectories from neural recordings [Sussillo and Abbott, 2009, Rajan et al., 2016, Pandarinath et al., 2018, Keshtkaran et al., 2022], to simulate circuit mechanisms of cognitive tasks [Yang et al., 2019, Driscoll et al., 2024], and to complement experiments through hypothesis generation [Pinto et al., 2019, Pagan et al., 2025]. In NeuroAI, RNNs have been embedded into deep reinforcement learning agents to recapitulate biological navigation codes of grid cells [Banino et al., 2018]. Yet, despite their prevalence, theoretical understanding of RNNs with biologically realistic, heavy-tailed weights remains limited.

To that end, the contribution of this paper is four-fold:

- We reveal that finite-size heavy-tailed RNNs exhibit a sharp transition from quiescence to chaos, *in contrast* to the mean-field prediction of ubiquitous chaos in infinite networks with `tanh`-like activation functions [Kuśmierz et al., 2020].

- We derive theoretical predictions for the critical gain at which this transition occurs as a function of network size, and validate them through simulations.

- We show numerically that heavier-tailed RNNs exhibit a slower transition to chaos, sustaining edge-of-chaos dynamics over a broader gain regime and offering greater robustness to gain variation; we show this can translate to improved information processing, as evidenced by the superior performance of heavy-tailed RNNs on a simple reservoir-computing task.

- We quantify attractor dimensionality as a function of tail heaviness, uncovering a tradeoff between robustness and dynamical complexity: heavier tails compress activity onto lower-dimensional manifolds.

## 2   Related Works

In his seminal work, Neal [1996] examined Bayesian inference in neural networks and demonstrated that, in the infinite-width limit, shallow feedforward networks with standard Gaussian weight initializations converge to Gaussian processes. He noted that this convergence breaks down when weights are drawn from Lévy $\alpha$-stable distributions, hypothesizing that such heavy-tailed initializations give rise to a richer class of priors beyond the representational capacity of Gaussian process kernels. This insight has since been extended and formalized by a recent series of theoretical works that rigorously

characterize the infinite-width limit of feedforward networks, showing convergence to $\alpha$-stable processes [Favaro et al., 2020, Jung et al., 2021, Bordino et al., 2023, Favaro et al., 2023]. Additionally, in [Favaro et al., 2024], the training dynamics of shallow feedforward networks with heavy-tailed distributions of weights are characterized through the neural tangent kernel [Jacot et al., 2018]. While these studies focus on feedforward architectures, our work complements them by uncovering and characterizing a distinct transition in heavy-tailed feedforward networks via an annealed analysis, an effect not previously reported. We then extend this investigation to recurrent networks.

The critical behavior of heavy-tailed networks has also been examined in both RNNs and feedforward settings. Wardak and Gong [2022] report an extended critical regime in heavy-tailed RNNs, while Qu et al. [2022] demonstrate that a similar extended critical regime emerges in heavy-tailed feedforward neural networks, with training via stochastic gradient descent being most efficient in the region of the parameter space corresponding to the critical regime. Our findings are consistent with these observations and advance them by: (a) explaining the extended critical regime in terms of the behavior of the maximal Lyapunov exponent, (b) showing that the location of the transition depends on the network size, and (c) identifying a tradeoff between the size of the critical regime and the dimensionality of the neural manifold in the critical regime.

Additionally, a mean-field theory of Cauchy RNNs (*i.e.,* with weights following a Lévy $\alpha$-stable distribution where $\alpha = 1$), is presented in Kuśmierz et al. [2020]. Specifically, they show that Cauchy RNNs with a binary activation function exhibit transition to chaos and generate scale-free avalanches, similarly observed in biological neural recordings [Beggs and Plenz, 2003] and often presented as evidence supporting the critical brain hypothesis [Muñoz, 2018]. Notably, Kuśmierz et al. [2020] note that Cauchy networks with a wide class of activation functions, including $\tanh$ studied in our work, are always chaotic in the infinite-size limit, and, as such, do *not* exhibit a transition to chaos. In contrast, our results reveal that this observation is no longer true in the *finite* networks, highlighting the importance of finite-size effects.

Finally, our work complements recent studies on brain-like learning with exponentiated gradients [Cornford et al., 2024], which showed that such updates naturally give rise to log-normal connectivity distributions. Within this broader context, our results offer a theoretical perspective that elucidates the dynamical consequences of these heavy-tailed structures.

## 3 Methods

### 3.1 Setup of recurrent neural network

We study recurrent neural networks that evolve in discrete time according to the update rule

$$x_i(t+1) = \phi\left(\sum_{j=1}^{N} W_{ij} x_j(t) + I_i(t)\right), \tag{1}$$

where $\phi = \tanh$ is the activation function, $I_i(t)$ is the external input to neuron $i$ at time $t$, and $N$ is the number of neurons. The synaptic weights $W_{ij}$ are independently drawn from a symmetric Lévy $\alpha$-stable distribution [Feller, 1971, Borak et al., 2005], i.e. $W_{ij} \sim L_\alpha(\sigma)$ with characteristic function

$$\phi_{L_\alpha(\sigma)}(k) = \exp\left(-|\sigma k|^\alpha\right), \tag{2}$$

and with scale parameter $\sigma = g/N^{1/\alpha}$, where the gain $g$ acts as the control parameter in our analysis. The stability parameter $\alpha \in (0, 2]$ affects the tails of the distribution. For $\alpha < 2$, the corresponding density function features heavy, power-law tails, i.e. $\rho_{L_\alpha(\sigma)}(x) \propto |x|^{-1-\alpha}$ when $|x| \gg 1$. The remaining case of $\alpha = 2$ corresponds to the familiar Gaussian distribution with light tails.

We perform analyses on both autonomous (zero-input) and stimulus-driven RNNs. In the latter case (see Appendix E), inputs at each time step are sampled i.i.d. from a Gaussian distribution with zero mean and variance $0.01$. This enables us to study how stochastic drive interacts with heavy-tailed synaptic weight distributions to modulate network stability.

### 3.2 Setup of feedforward networks

Although the weight matrix remains fixed during RNN evolution, in our mathematical analysis, we assume that $W_{ij}$ is redrawn at each time step. With such an *annealed* approximation [Derrida and

Pomeau, 1986], evolving the RNN for $T$ steps effectively corresponds to passing an input (initial condition) through a feedforward network of $T$ layers. In this case, we can reformulate the update equation (1) as

$$x_i^{(t+1)} = \phi \left( \sum_{j=1}^{N_t} W_{ij}^{(t)} x_j^{(t)} + I_i^{(t)} \right), \tag{3}$$

where $W_{ij}^{(t)}$ is the $N_{t+1} \times N_t$ weight matrix at layer $t$. In this case, the initial condition $x_i^{(0)}$ is interpreted as the input, and activity at $t = T$ as the output of a $T$-layer network. Additional inputs could also be passed directly to each layer via $I_i^{(t)}$. Note that we assumed that each layer may have a different width $N_t$. The case when $\forall_t N_t = N$ corresponds to the annealed approximation of (1). We use $\langle \cdot \rangle_X$ to denote the expected value with respect to a random variable $X$.

### 3.3 Computation of Lyapunov exponents

To quantify the dynamical stability of RNNs, we compute their Lyapunov exponents across a range of weight scales (gains) $g$. This also provides an estimate of the maximum Lyapunov exponent (MLE, $\lambda_{\mathrm{max}}$), which measures the average exponential rate at which nearby trajectories diverge in phase space. A positive $\lambda_{\mathrm{max}}$ indicates chaotic dynamics, while a negative value implies convergence to a stable fixed point or limit cycle. When $\lambda_{\mathrm{max}} \approx 0$, the system operates at the *edge of chaos*, a critical regime where perturbations neither grow nor decay rapidly.

We adopt the standard QR-based algorithm [Von Bremen et al., 1997] described in Vogt et al. [2022] (detailed in Appendix C) to compute the Lyapunov spectrum. For each input sequence, we track how infinitesimal perturbations evolve under the hidden-state Jacobians. These perturbations are orthonormalized via QR decomposition at each step, and the logarithms of the diagonal entries of the $R$ matrix are accumulated to estimate the exponents. To avoid transient effects, we include a short `warmup` period during which the network state evolves but Lyapunov exponents are not accumulated. The MLE is then averaged over multiple random input sequences to obtain a robust estimate.

### 3.4 Participation ratio and Lyapunov dimension

**Two notions of dimensionality** We analyze the dimensionality of RNN dynamics from two complementary perspectives. The first, based on Lyapunov exponents, quantifies how many directions exhibit local expansion under small perturbations; this is captured by the Lyapunov (Kaplan–Yorke) dimension $D_{\mathrm{KY}}$, derived from the leading part of the Lyapunov spectrum. The second, based on the participation ratio (PR), measures how many orthogonal directions the network activity spans at steady state, using second-order statistics of the hidden states. Intuitively, PR is a linear method that approximates the manifold by an ellipsoid, and as such it may significantly overestimate the dimensionality of a highly nonlinear manifold. In contrast, $D_{\mathrm{KY}}$ is a nonlinear measure that, for typical systems, correctly estimates the information (fractal) dimension of a chaotic attractor [Ott, 2002].

**Lyapunov dimension** To measure intrinsic dynamical complexity, we compute the full Lyapunov spectrum using the standard QR method (see Section 3.3). Let $\lambda_1 \geq \lambda_2 \geq \cdots \geq \lambda_N$ be the ordered Lyapunov exponents. Define $k$ as the largest index such that $\sum_{i=1}^{k} \lambda_i \geq 0$. Then the Lyapunov dimension [Frederickson et al., 1983, Farmer et al., 1983, Ott, 2002] is defined as:

$$D_{\mathrm{KY}} := k + \frac{\sum_{i=1}^{k} \lambda_i}{|\lambda_{k+1}|}. \tag{4}$$

Near the edge of chaos, all positive Lyapunov exponents are close to 0 and perturbations along the corresponding directions expand with slow timescales. As a result, in this regime, a higher $D_{\mathrm{KY}}$ indicates that the system evolves on a higher-dimensional *slow manifold* [Krishnamurthy et al., 2022], with more modes contributing to long-term variability and slow divergence–implying a greater capacity to support rich, temporally extended computations. We track all orthogonal directions and update them with QR decomposition at each step after a fixed `warmup`. We examine how $D_{\mathrm{KY}}$ evolves with gain $g$ across all dynamical regimes.

**Participation ratio** Let $x^{(t)} \in \mathbb{R}^N$ be the hidden state of the RNN at time $t$, recorded over the final $K$ steps of a length-$T$ trajectory at fixed gain $g$ with $K > N$, after discarding the initial $T - K$ `warmup` steps. We compute the empirical covariance matrix $S = \frac{1}{T-1} \sum_{t=1}^{T} \left( x^{(t)} - \bar{x} \right) \left( x^{(t)} - \bar{x} \right)^{\top}$, where $\bar{x} = \frac{1}{T} \sum_{t=1}^{T} x^{(t)}$. Let $\tilde{\lambda}$ denote the eigenvalues of $S$. The participation ratio is defined as [Kramer and MacKinnon, 1993, Gao et al., 2017, Recanatesi et al., 2022]:

$$\mathrm{PR} := \frac{\left( \sum_i \tilde{\lambda}_i \right)^2}{\sum_i \tilde{\lambda}_i^2}. \tag{5}$$

PR ranges from 1 (all variance in one mode) to $N$ (uniform variance), and quantifies how many orthogonal directions carry substantial variance regardless of stability. It has been widely used to characterize neural dimensionality in biological and artificial circuits [Gao et al., 2017, Recanatesi et al., 2022]. We compute PR across all types of regimes for the post-`warmup` steady-state trajectories.

## 4 Results

### 4.1 Finite heavy-tailed networks exhibit a predictable quiescent-to-chaotic transition

#### 4.1.1 Information propagation in feedforward networks

We study networks without external inputs ($I^{(t)} = 0$). Since $\phi(0) = 0$, the quiescent state is a fixed point of both (1) and (3). In our mathematical derivation, we focus on the simpler case of annealed dynamics. To study the stability of the quiescent state, we expand (3) around $x^{(t)} = 0$ and obtain a linear equation

$$\varepsilon^{(t+1)} = W^{(t)} \varepsilon^{(t)} \tag{6}$$

where we used $\phi'(0) = 1$. Since sequences of weights at successive layers are generated i.i.d., (6) corresponds to the Kesten process [Kesten, 1973]. When $t \to \infty$, the Kesten process may either converge to a limiting distribution or diverge. In our case, the width of the distribution of entries of $W^{(t)}$ acts as a parameter that controls the transition between these two qualitatively distinct behaviors.

A detailed analysis in Appendix A shows that the critical width of the distribution is given by

$$g^* = \exp\left( -\langle \Xi_{N,\alpha} \rangle \right) \tag{7a}$$

$$\Xi_{N,\alpha} = \frac{1}{\alpha} \ln \left( \frac{1}{N} \sum_{j=1}^{N} |z_j|^\alpha \right) \tag{7b}$$

with $z_j \sim L_\alpha(1)$. Let us first show that this formula is consistent with the known results in the Gaussian case. Noting that in our notation for $\alpha = 2$ we have $\langle z^2 \rangle = 2$, we can take the limit $N \to \infty$ and obtain $\Xi_{N \to \infty, 2} = \ln \sqrt{2}$. This leads to $g^* = 1/\sqrt{2}$ and $L_2(g^*) \sim \mathcal{N}(0, 1)$. Hence, we recover the well-known transition at $\langle W_{ij}^2 \rangle = 1/N$ [Sompolinsky et al., 1988, Molgedey et al., 1992, Toyoizumi and Abbott, 2011]. Our formula, however, is more general and applies to any finite width of the network. It predicts that $g^*$, for any fixed $\alpha$, is a decreasing function of $N$ (Fig. 1A). In the Gaussian case, it quickly reaches its asymptotic value consistent with the mean-field prediction. In heavy-tailed networks, however, the decay is slow and is clearly visible across four orders of magnitude shown in Fig. 1A. Our theory predicts that this decay is logarithmic with an $\alpha$-dependent exponent, i.e., $g^* \propto 1/(\ln N)^{1/\alpha}$ for $\alpha < 2$, see Appendix B for the derivation.

We also confirm our theoretical predictions in simulations, by passing a random initial vector through $T = 100$ steps (layers) of a linearized network with weights redrawn at each step from a fixed distribution. We fix $\alpha = 1$ and vary $g$. Below (above) the transition, we expect the components of the final state to be close to (far from) zero with high probability. Thus, we construct a simple order parameter $f_{<\epsilon}$ defined as the number of components of the final state $\varepsilon(T)$ that are within $\epsilon$ from 0. As shown in Fig. 1B, the network goes through a sharp transition between $f_{<\epsilon} = 1$ and $f_{<\epsilon} = 0$, and the location of the transition is consistent with our theoretical prediction. The transition is rather sharp even for small networks (Fig. 1C, $N = 100$), and is expected to become even sharper with increasing number of steps $T$. Our theoretical result becomes exact in the limit of $T \to \infty$.

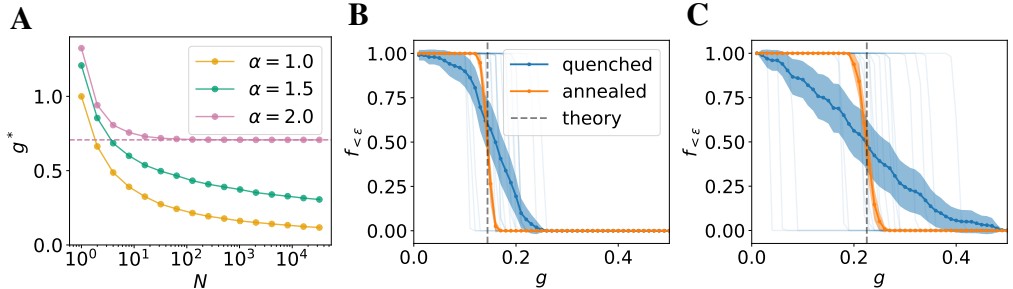

Figure 1: **(A)**: Transition point $g^*$ predicted by our theory as a function of network size for various $\alpha$. The transition point of Gaussian networks rapidly converges to the mean-field limit (dashed line). In contrast, the transition point of heavy-tailed networks decays slowly towards zero. **(B)**: The fraction of small ($\epsilon = 0.1$) final state components in linear networks with $\alpha = 1$ and $N = 3000$ evolved for $T = 100$ steps from random initial conditions as a function of $g$. In the annealed case, we observe a sharp transition at the location predicted by the theory. In the quenched case, each individual realization exhibits a sharp transition (thin blue lines), but its location varies between different realizations of the weight matrix. Thus, when averaged over the realizations (thick blue line and dots; shaded region shows the $\pm 3$ standard error), the transition looks smoother than in the annealed case. Nonetheless, its location is approximately predicted by the theory. **(C)**: Same as B but with $N = 100$. As predicted by the theory, the transition point shifts to the right with decreasing $N$. Moreover, the location of the transition in the quenched case varies more in smaller networks.

### 4.1.2 Quenched disorder and recurrent neural networks

In contrast to our annealed analysis of feedforward networks, the weights of the RNN remain constant throughout the evolution. Since we are interested in finite-sized networks, we can expect the location of the transition to vary between the realizations of the weight matrix. A mathematical analysis of this phenomenon is beyond the scope of this work. We expect, however, that the random fluctuations of $g^*$ in networks with quenched random weights should be concentrated around the annealed prediction and should decrease with $N$. Moreover, the typical values of $g^*$ should decrease with $N$ as predicted by the annealed theory.

To test this hypothesis, we simulate a quenched version of (6) in which $\forall_t W^{(t)} = W$. In order to observe the transition, we first fix the random seed (i.e., draw random components of the weight matrix from $L_\alpha(1/N^{1/\alpha})$) and then rescale its components by various values of $g$. We focus on $\alpha = 1$ as a representative example. As shown in Fig. 1B, evolution of each realization of the weight matrix goes through a very sharp transition, but the location of this transition varies significantly between the realizations. Nonetheless, they are concentrated around the point predicted by the annealed theory. Moreover, the location of the transition shifts to the right and fluctuations increase with decreasing $N$ and (Fig. 1C). These results suggest that the location of the transition in the quenched case approaches the annealed prediction with increasing $N$. We provide further analysis on the behavior of the quanched transition point as a function of network size $N$ in Appendix J.

Note that our theoretical analysis does not specify the nature of the dynamics above the transition. Our simulations indicate that the network hovers around the edge of chaos in a significant range of values of $g$ and, for $\alpha \geq 1$, ultimately enters chaotic regime (see Fig. 2). In contrast, the dynamics of networks with $\alpha < 1$ show a non-monotonic behavior of the MLE: after staying near the edge of chaos at intermediate values of $g$, the dynamics seem to ultimately settle in a stable, non-chaotic regime at larger values of $g$ (see Appendix D). Thus, for $\alpha \geq 1$, our annealed prediction $g^*$ gives the approximate location of the transition to chaos in RNNs. Although our analysis focused on autonomous dynamics, similar to the Gaussian case, we expect noise to shift, but not completely remove, the transition [Molgedey et al., 1992, Rajan et al., 2010].

While our analysis focuses on the `tanh` activation, it generalizes to any function satisfying $\phi(0) = 0$ and admitting a local expansion $\phi(x) = ax + o(x)$. In this regime, the existence of the transition follows directly from the linear stability of the quiescent fixed point. For unbounded activations such as `ReLU`, however, bounded dynamics are no longer guaranteed, and divergence may occur at large $g$. Although the transition itself persists for a broad class of activations, the qualitative behavior above it can differ substantially. Beyond the transition, the ensuing dynamics depend sensitively on the nonlinearity: linear or `ReLU` activations typically diverge for large $g$, whereas sublinear, saturating

nonlinearities constrain activity and preserve stability. We therefore expect our results to hold for any smooth, saturating activation function, while unbounded ones likely produce more complex, divergent dynamics. The framework can also be extended to cases with $\phi(0) \neq 0$ by expanding around the corresponding non-quiescent fixed point. Here, the fixed point's location may vary with the order parameter $g$, but the overall nature of the transition should remain unchanged.

## 4.2 Heavier-tailed RNNs exhibit a slower, more robust transition to chaos

Having established the existence of a finite-size transition between quiescent and chaotic dynamics in RNNs with heavy-tailed synaptic weights (Section 4.1), we next examine how the nature of this transition differs across tail indices $\alpha$. Our simulations of autonomous RNNs (Fig. 2; similar results for noisy stimulus-driven RNNs shown in Fig. 5) reveal that although networks with $\alpha \geq 1$ exhibit a transition to chaos as predicted, the sharpness and location of the transition vary substantially with $\alpha$, in which a lower value corresponds to a heavier-tailed distribution.

In networks with Gaussian connectivity ($\alpha = 2.0$), the maximal Lyapunov exponent (MLE) increases steeply with gain $g$, indicating a rapid onset of chaos. In contrast, RNNs with heavier-tailed weights (lower $\alpha$) exhibit a slower rise in the MLE as $g$ increases near the transition (when the MLE is near zero). This gradual transition implies that these networks remain closer to the edge of chaos over a wider range of gain values, consistent with previous observations of an extended, critical-like region [Wardak and Gong, 2022, Qu et al., 2022]. Such extended critical-like behavior can offer a form of robustness with respect to changes in network parameters, which can be an important property that benefits biological networks in non-stationary environments, allowing the network to maintain sensitive high-capacity dynamics [Bertschinger et al., 2004, Legenstein and Maass, 2007, Toyoizumi and Abbott, 2011] without the requirement of precise parameter adjustment. In our analysis of reservoir-computing networks on a delayed XOR task (Appendix L), heavy-tailed networks maintained strong task performance across a broader gain regime than Gaussian networks. This provides a concrete proof-of-concept that the extended critical regime enhances robustness and performance without fine-tuning, which may benefit both machine learning applications and neural computation. Future studies could extend this analysis to trained recurrent networks and more complex temporal tasks to further elucidate how heavy-tailed connectivity shapes information processing and learning dynamics.

Moreover, as shown in Fig. 2 (with $N$ increases from left to right panels), the locations of the transition (MLE $\geq 0$) shift to the left (lower $g$) as $N$ increases. Notably, this shift is more pronounced in heavier-tailed networks. This finding echoes our theoretical prediction that the critical gain $g^*$ slowly decreases with increasing $N$ in the heavy-tailed regime due to the finite-size effect (Fig. 1A).

Together, these simulation results provide empirical evidence that while infinite-width mean-field theory predicts ubiquitous chaos for Lévy RNNs, finite-size networks can operate near a well-defined, robust transition point whose properties depend systematically on the tail index $\alpha$ and network size $N$. This behavior may be particularly relevant in biological systems, where recent experimental evidence suggests synaptic weights follow heavy-tailed statistics, and where robustness to parameter variation is essential. Our findings imply that heavy-tailed connectivity may naturally support computations at the edge of chaos in finite-size neural circuits without requiring fine-tuning.

## 4.3 Heavy-tailed RNNs compress the chaotic attractor into a lower-dimensional slow manifold

The robustness of transition to chaos in heavy-tailed RNNs raises a natural question: does the structure of the underlying dynamical landscape also vary systematically with respect to the tail index $\alpha$?

To address this, we first examine the full Lyapunov spectrum of networks near the transition to chaos, then we further characterize the effective dimensionality of the network's dynamics using two complementary metrics: the Lyapunov dimension ($D_{\mathrm{KY}}$), which estimates how many directions in phase space are locally expanding or marginally stable [Frederickson et al., 1983, Farmer et al., 1983, Ott, 2002], and the participation ratio (PR), which captures how variance is distributed across neural population activity and is commonly used in neuroscience [Kramer and MacKinnon, 1993, Gao et al., 2017, Recanatesi et al., 2022]. We find that although heavy-tailed networks benefit from robustness near the edge of chaos, this comes with a key tradeoff: the dynamics are compressed into a lower-dimensional slow manifold.

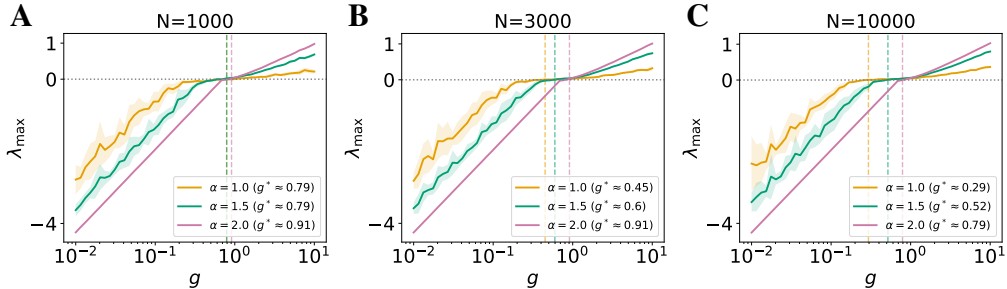

Figure 2: **Maximum Lyapunov exponent ($\lambda_{\mathrm{max}}$) as a function of gain $g$ for autonomous recurrent networks with different tail indices $\alpha$, shown for**: **(A)** $N = 1000$, **(B)** $N = 3000$, and **(C)** $N = 10000$. Curves show mean across 10 trials; shaded regions denote $\pm 1$ SD. We let the networks evolve for $T = 3000$ steps, among which the Lyapunov exponents are accumulated over the last $K = 100$ steps. See results under noisy stimulus and ablation studies in Appendices E, G. Heavier-tailed networks (lower $\alpha$) exhibit a slower, more gradual increase in $\lambda_{\mathrm{max}}$ near the transition (where $\lambda_{\mathrm{max}} = 0$), resulting in a broader edge-of-chaos regime with respect to $g$. Dashed lines and legend mark the average critical gain $g^*$ at which $\lambda_{\mathrm{max}}$ first crosses zero. As $N$ increases, this transition shifts leftward, especially for lower $\alpha$, in line with our theoretical predictions on finite-size effects.

### 4.3.1 Lyapunov spectrum shows compressed slow manifold in heavy-tailed RNNs

To probe the structure of the dynamical landscape near the transition to chaos, we examine the full Lyapunov spectrum of the networks. The spectrum provides a detailed view of local stability across phase space, with each Lyapunov exponent characterizing the growth or decay of perturbations along a particular direction in the network's state space. In particular, the density of the exponents near zero reflects the presence of a slow activity manifold where the network evolves in the steady state. The slow manifold contains marginally stable modes along which input-driven perturbations expand or shrink slowly. Thus, in the absence of other memory mechanisms, slow modes endow RNNs with a crucial capacity to integrate information across long timescales [Krishnamurthy et al., 2022].

In Fig. 3A, we show the Lyapunov spectra for Gaussian and heavy-tailed networks near their respective estimated critical gain when the MLE first exceeds zero. The average critical gain $\langle g^* \rangle$ is estimated through ten realizations in Fig. 2A and the histograms are averaged across runs with the same $\langle g^* \rangle$, hence they contain positive Lyapunov exponents (see Appendix F for individual realizations across input conditions, and additional discussion on the overestimation of $\langle g^* \rangle$). The distribution of Lyapunov exponents differs markedly between these two types of network connectivity. Gaussian networks show a dense band of exponents concentrated near zero, indicating a broad, slow manifold. In contrast, as $\alpha$ decreases (*i.e.,* the heaviness of the tail increases), the number of Lyapunov exponents near zero decreases, revealing a *compression* of the slow manifold.

This suggests a tradeoff between the robustness of the edge of chaos and the dimensionality of the slow manifold. Following this observation, we next quantitatively characterize the attractor dimensionality.

### 4.3.2 Lyapunov dimensions and participation ratio further characterize low attractor dimensionality in heavy-tailed RNNs

To further characterize the tradeoff introduced by heavy-tailed connectivity, we quantify the dimensionality of the dynamical attractor using two complementary metrics (detailed in Section 3.4).

First, we compute the Lyapunov dimension ($D_{\mathrm{KY}}$), which estimates the effective number of directions in phase space that exhibit local expansion [Frederickson et al., 1983, Farmer et al., 1983, Ott, 2002]. This measure reflects the intrinsic complexity of the system's attractor. As shown in Fig. 3B, recurrent networks with heavier-tailed synaptic weights (lower $\alpha$) exhibit a significantly lower $D_{\mathrm{KY}}$ than their Gaussian counterparts across the near-chaotic regime (characterized in Fig. 2). This confirms that despite their robustness to chaos, heavy-tailed networks operate on lower-dimensional attractors.

Second, we evaluate the participation ratio (PR), a widely used metric for gauging the effective dimensionality of neural population activity. It has been leveraged to quantify task-relevant low-dimensional subspaces and other properties of multi-unit neuronal recordings in behaving animals

[Gao et al., 2017], and summarize the collective modes visited by recurrent spiking networks and reveal how these modes depend on local connectivity motifs [Recanatesi et al., 2019]. PR measures how variance in population activity is distributed across the eigenmodes of the covariance matrix, providing a compact read-out of the number of degrees of freedom the network explores [Kramer and MacKinnon, 1993]. As shown in Fig. 3C, PR declines as $\alpha$ decreases, although the drop is shallower than that of $D_{KY}$. This difference is expected: PR is a second-order statistic that is sensitive to how variance is spread across modes, whereas $D_{KY}$ is a quantity set by local expansion rates of the flow. Consequently, PR can remain relatively high even when only a few directions in phase space are truly unstable, highlighting complementary information provided by these two dimensionality measures.

We hypothesize that the large disparity in Lyapunov dimensions between Gaussian and heavy-tailed networks arises from the broader dispersion of Lyapunov exponents in the latter as shown in Fig. 3A. Intuitively, only a small subset of leading exponents becomes positive near the edge of chaos in heavy-tailed networks, resulting in a lower overall Lyapunov dimension. This effect likely reflects the more heterogeneous eigenvalue distribution of the underlying weight matrix. However, the precise mapping between the weight matrix spectrum and the Jacobian's Lyapunov spectrum remains nontrivial and warrants further analysis.

Together, these metrics reveal that the slow manifold in heavy-tailed RNNs is both more contractive (lower $D_{KY}$) and narrower (lower PR), supporting the view that these networks "prioritize" robustness over dynamical richness. This tradeoff is biologically aligned with observations in animal studies, where low-dimensional neural representations are often found relative to the high-dimensional ambient space of neural recordings, even in complex behaviors [Nieh et al., 2021, Cueva et al., 2020, Chaudhuri et al., 2019, Yoon et al., 2013]. We return to this point in the Discussion.

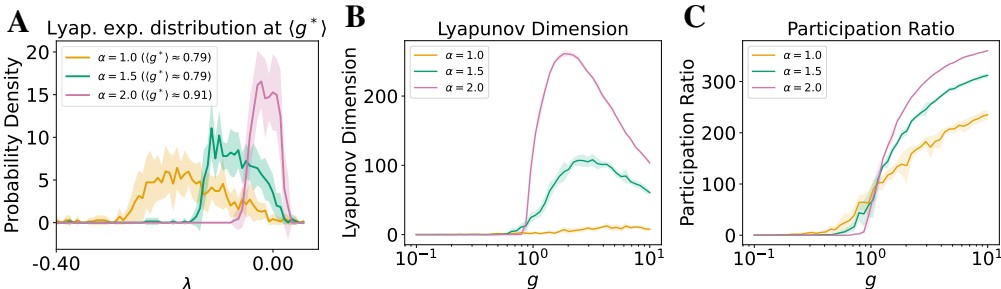

Figure 3: **Heavy-tailed networks exhibit lower-dimensional attractors near the edge of chaos.** Curves show mean across 10 trials for networks of size $N = 1000$; shaded regions denote $\pm 1$ SD. See Appendix E for results under noisy stimuli. The implementation details and ablation studies are provided in Appendices H, I. (**A**) Distributions of top 100 Lyapunov exponents for varying $\alpha$ show fewer exponents near zero in heavier-tailed networks at estimated $\langle g^* \rangle$ obtained in Fig 2A, indicating a lower-dimensional slow manifold. x-axis truncated at the left to omit near-zero tails for clarity. (**B**) The Lyapunov dimension is smaller for heavier-tailed networks near the regime of edge of chaos, reflecting fewer directions of local expansion in phase space. (**C**) The participation ratio dimension is similarly smaller with lower $\alpha$ near the edge of chaos, showing reduced variance homogeneity across neural modes. Together, these results indicate that while heavy-tailed networks maintain robustness to neural gain near chaos, they compress dynamics into a lower-dimensional attractor.

## 5 Discussion

Critically, our findings are with respect to finite-size networks and depend on network size. In the infinite-width limit, mean-field theory predicts that Lévy networks are always chaotic (Section 4.1.1). However, our results show that finite-size networks exhibit a clear quiescent-to-chaotic transition, with the critical gain $g^*$ shifting systematically with both network size $N$ and tail index $\alpha$ (Eqn. 7 and Fig. 2). This highlights that mean-field approximations may miss important structure in biologically sized circuits, and that finite-size corrections offer a more accurate theoretical framework for understanding real neural systems that are finite in size.

Further, as demonstrated in Fig. 2, heavy-tailed weight distributions make RNNs more robust to changes in gain, a parameter that may correspond biologically to either the width of synaptic weight distributions or to neural gain modulated by neuromodulatory systems [Waterhouse et al., 1988, Shine

et al., 2018]. Specifically, we observe that networks with heavier-tailed synaptic weights remain near the edge of chaos over a much wider range of gain values than those with Gaussian connectivity, which is commonly assumed in theoretical studies. This property may be especially valuable for biological systems that operate across multiple states (e.g., sleep and waking [Chaudhuri et al., 2019]) or in non-stationary environments. Such robustness could help explain empirical findings that similar neural activity patterns can arise from vastly different underlying circuit parameters in healthy brains [Prinz et al., 2004, Marder, 2011]. Meanwhile, as shown in Fig. 3, heavier tails reduce both the Lyapunov dimension and the participation ratio, indicating that the slow manifold supporting long-lasting activity becomes lower-dimensional. Our further analysis show that a handful of "mega-synapses" drives the dynamics, implying the robustness and low-dimensionality largely stem from extreme outliers (Appendix K). Together, these effects imply a tradeoff: heavy-tailed networks are more stable to perturbations but require more neurons to achieve the same computational capacity, such as memory or temporal integration, compared to Gaussian networks.

Notably, a common empirical observation in neuroscience is that neural population activity tends to evolve within a low-dimensional manifold relative to the large number of neurons recorded. This phenomenon has been observed across cortical and subcortical regions, and is often behaviorally meaningful [Nieh et al., 2021, Bondanelli et al., 2021]. Theoretical work suggests that low-dimensionality can arise from constraints imposed by circuit connectivity [Mastrogiuseppe and Ostojic, 2018] or task demands [Gao et al., 2017]. Our finding that heavier-tailed RNNs yield lower-dimensional attractors biologically aligns with this widespread phenomenon and provides evidence that anatomical connectivity might constrain the expressive capacity of population activity.

The observed robustness-dimensionality tradeoff also offers predictions for which tasks heavy-tailed circuits can be best suited. Tasks requiring only low-rank dynamics for reliable integration or pattern generation (*e.g.,* binary decision-making [Brunton et al., 2013] or working memory [Panichello and Buschman, 2021]) may benefit from the extended edge-of-chaos regime provided by heavy-tailed weights. In contrast, tasks that rely on high-dimensional dynamics—such as representing multiple independent memories or generating complex trajectories (*e.g.,* virtual reality navigation [Busch et al., 2024])—may require larger networks or connectivity distributions closer to Gaussian. These predictions can be tested using emerging connectomic [Dorkenwald et al., 2024, The MICrONS Consortium, 2025] and large-scale recording datasets [Bondy et al., 2024, Manley et al., 2024], which can jointly measure synaptic weight distributions and task-related activity dimensionality.

Our framework can be naturally extended to a mixture setting, improving biological plausibility. For instance, neurons could be homogeneous and each draw weights randomly from one of multiple heavy-tailed distributions or form interacting subpopulations with distinct $\alpha$ values and connectivity motifs. Such extensions may capture diversity across neuronal cell types [Jin et al., 2025, Zeng, 2022] and offer a promising direction for future work.

We acknowledge several limitations: our study used untrained rate-based networks with homogeneous units. Including more biologically realistic features such as spiking dynamics [Kim et al., 2019], Dale's law [Dale, 1935], cell-type diversity [Zeng, 2022, Yao et al., 2023], and synaptic plasticity [Citri and Malenka, 2008] could modify or refine the observed effects. Furthermore, while our results focused on untrained dynamics, a key next step is to study how learning algorithms interact with the broad critical regime and how trained or reservoir computing heavy-tailed networks perform across a range of tasks [Yang et al., 2019, Driscoll et al., 2024]. Such a study would help us to understand and predict, based on connectivity alone, what kinds of computations a brain-like circuit is suited to perform—an important goal as we seek to interpret rich new connectomic datasets and understand how synaptic connectivity ties to function [Garner et al., 2024, Seung, 2024]. Another valuable next step is to extend this work toward direct comparison with neural recordings. For example, future studies could estimate Lyapunov spectra or related dynamical signatures from long, high-resolution neural activity trajectories. While such analyses are technically challenging and require stable, extended recordings, they would offer a powerful bridge between theory and experiment.

In summary, finite-size recurrent networks with previously understudied Lévy-distributed weights reveal a clear rule: heavier-tailed synaptic connectivity widens the regime of stable, edge-of-chaos dynamics but reduces the dimensionality of the resulting activity. This tradeoff links connectivity statistics, network size, and functional capacity, offering a principled, biologically plausible framework for interpreting both biological data and designing more parameter-robust artificial systems.

# 6   Acknowledgements

Eva Yi Xie was supported by a NeurIPS Foundation travel award for the NeurIPS conference presentation. Support for Stefan Mihalas was provided in part by NSF (2223725) and NIH (RF1DA055669); support for Łukasz Kuśmierz was provided in part by NSF (2223725). We are grateful to the colleagues at the Allen Institute and the COSYNE 2025 community for feedback on an early version of this work.

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

# A  Mathematical analysis of the transition in annealed networks

Our goal is to show that networks with $\alpha$-stable weight distributions exhibit a transition between two regimes, and to find the location of this transition which we denote as $g^*$. As in Gaussian networks, the quiescent state is stable and any small perturbation around it shrinks if weights are generated from a narrow enough distribution (i.e., $g < g^*$). Similarly, the quiescent state is unstable if the underlying distribution is wide enough ($g > g^*$). In contrast to Gaussian networks, however, this effect can only be observed through the analysis of finite-size effects.

As described in the main text, we study linear stability of the quiescent fixed point of (6). Since weights are randomly redrawn at each step, the evolution $\varepsilon^{(t)}$ is a stochastic process. To quantify its behavior we focus our attention on the conditional distribution $\varepsilon^{(t+1)}$ given $\varepsilon^{(t)}$. Components of this vector are independent due to the assumed independence of rows of the weight matrix. The conditional distribution of a single component can be characterized in the Fourier space as

$$\left\langle \exp\left(ik\varepsilon_i^{(t+1)}|\varepsilon^{(t)}\right)\right\rangle_W = \left\langle \exp\left(ik\sum_{j=1}^{N_t} W_{ij}^{(t)}\varepsilon_j^{(t)}\right)\right\rangle_{W_{ij}^{(t)}} = \exp\left(-|k|^\alpha g^\alpha \frac{1}{N_l}\sum_{j=1}^{N_t}\left|\varepsilon_j^{(t)}\right|^\alpha\right) \tag{8}$$

where we used $W_{ij}^{(t)} \sim L_\alpha\left(g/N_t^{1/\alpha}\right)$. Thus, for $t > 1$ the perturbation, when conditioned on the previous step, is an $\alpha$-stable random variable. More specifically, it can be written as $\varepsilon_i^{(t+1)}|\varepsilon^{(t)} \sim L_\alpha\left(\gamma^{(t+1)}\right)$, where the conditional scale at step $t+1$

$$\gamma^{(t+1)} = g\left(\frac{1}{N_t}\sum_{j=1}^{N_t}\left|\varepsilon_j^{(t)}\right|^\alpha\right)^{1/\alpha} \tag{9}$$

is a deterministic function of state at time $t$, which itself is a random variable. We can unpack this relation one step backwards by conditioning on $\varepsilon^{(t-1)}$ instead, with $\varepsilon_i^{(t)}|\varepsilon^{(t-1)} \sim L_\alpha\left(\gamma^{(t)}\right)$. We utilize the fact that this can also be expressed as

$$\varepsilon_i^{(t)}|\varepsilon^{(t-1)} = \gamma^{(t)}z_i^{(t)} \tag{10}$$

where $\gamma^{(t)}$ depends on the perturbation at time $t-1$, and $z_i^{(t)}$ are i.i.d. $\alpha$-stable variables. This leads to the recursive formula for scalar $\gamma^{(t)}$

$$\gamma^{(t+1)} = \gamma^{(t)}\xi^{(t)} \tag{11}$$

where $\left(\xi^{(t)}\right)_{t=1}^\infty$ is a sequence of independent random variables distributed as

$$\xi^{(t)} = g\left(\frac{1}{N_t}\sum_{j=1}^{N_t}\left|z_j^{(t)}\right|^\alpha\right)^{1/\alpha} \tag{12}$$

with i.i.d. $z_j^{(l)} \sim L_\alpha(1)$. If layers have the same width $N_t = N$, $\xi^{(t)}$ are i.i.d. and (11) is a scalar multiplicative process with i.i.d. entries. Thus, we have reduced our problem to a simpler special case of purely multiplicative scalar Kesten process. We can easily solve this recursion and rewrite the solution as a sum

$$\ln\gamma^{(t+1)} = \ln\gamma^{(t)} + \sum_{i=1}^t \ln\xi^{(i)} \tag{13}$$

where $\gamma^{(1)}$ is deterministically specified by the input perturbation $\varepsilon^{(0)}$. It is known [Kesten, 1973, Statman et al., 2014] that this sum diverges to $-\infty$ almost surely if $\langle\ln\xi\rangle < 0$ and diverges to $\infty$ almost surely if $\langle\ln\xi\rangle > 0$. Accordingly, the sequence $\left(\gamma^{(t)}\right)_{t=1}^\infty$ either converges to 0 or diverges. Therefore, the critical width of the synaptic weight distribution is given by

$$g^* = \exp\left(-\langle\Xi_{N,\alpha}\rangle\right) \tag{14}$$

where

$$\Xi_{N,\alpha} = \frac{1}{\alpha}\ln\left(\frac{1}{N}\sum_{j=1}^N |z_j|^\alpha\right) \tag{15}$$

with $z_j \sim L_\alpha(1)$.

# B   Derivation of the logarithmic decay of $g^*(N)$

Here, we estimate the expected value of

$$\Xi_{N,\alpha} = \frac{1}{\alpha} \ln \left( \frac{1}{N} \sum_{j=1}^{N} |z_j|^\alpha \right), \tag{16}$$

where $z_j \sim L_\alpha(1)$, for large $N$. We define $Y_{N,\alpha} = \frac{1}{N} \sum_{j=1}^{N} |z_j|^\alpha$ and note that the Laplace transform of $Y_{N,\alpha}$ can be calculated as

$$F_{N,\alpha}(s) = \left\langle e^{-sY_{N,\alpha}} \right\rangle = \left( F_{1,\alpha} \left( \frac{s}{N} \right) \right)^N \tag{17}$$

where

$$F_{1,\alpha}(s) = \left\langle e^{-s|z|^\alpha} \right\rangle_{z \sim L_\alpha(1)} \tag{18}$$

According to (17), the large $N$ asymptotic of $\Xi_{N,\alpha}$ is dominated by the behavior of $F_{1,\alpha}(s)$ around $s = 0$. This behavior should be similar for all symmetric distributions with the same stability index. For example, take $\rho_z(x) = \frac{\alpha}{2} |x|^{-1-\alpha}$ for $|x| > 1$ and $\rho_z(x) = 0$ otherwise. The resulting expansion can be found as

$$\left\langle e^{-s|x|^\alpha} \right\rangle_{z \sim \rho_z} = s \int_s^\infty du\, u^{-2} e^{-u} = s\Gamma(-1, s) \approx 1 - s\left(1 - \gamma - \ln s\right) + O(s^2) \tag{19}$$

where $\Gamma(a, s)$ is the upper incomplete gamma function and $\gamma$ is the Euler-Mascheroni constant. Thus, the asymptotic expansion of $F_{1,\alpha}(s)$ must take the form

$$F_{1,\alpha}(s) = 1 - A_\alpha s \left(B_\alpha - \ln s\right) + O(s^2) \tag{20}$$

for some irrelevant constants $A_\alpha, B_\alpha$. We plug (20) into (17) and arrive at

$$\ln F_{N,\alpha}(s) = -A_\alpha s \left(B_\alpha - \ln s + \ln N\right) + O\left(N^{-1}\right) \tag{21}$$

For $N \gg 1$, (21) corresponds to a random variable $X_N$ that can be constructed as

$$X_N = X_1 + A_\alpha \ln N, \tag{22}$$

where

$$\langle \exp(-sX_1) \rangle = \exp(-A_\alpha s \left(B_\alpha - \ln s\right)) \tag{23}$$

We can rewrite the desired expected value as

$$\langle \Xi_{N,\alpha} \rangle \approx \frac{1}{\alpha} \left\langle \ln \left(X_1 + A_\alpha \ln N\right) \right\rangle_{X_1} \tag{24}$$

The distribution of $X_1$ is fixed and does not change with $N$. Thus, for large $N$ the second term dominates, and we arrive at

$$g^* = \exp(-\langle \Xi_{N,\alpha} \rangle) \asymp \frac{1}{(\ln N)^{1/\alpha}} \tag{25}$$

# C Algorithm to compute Lyapunov exponents for RNNs

We leverage the algorithm proposed in Vogt et al. [2022] to study the dynamics of RNNs, adapted to our setting where we process a single input sequence at a time (*i.e.*, batch size = 1). Then, running multiple realizations simply means running the same algorithm but with a different seed set in the beginning; this is equivalent to having a batch of inputs shown in the original algorithm in Vogt et al. [2022].

To reduce the influence of transient dynamics, we include a `warmup` period during which the RNN is evolved forward but Lyapunov exponents are not yet accumulated.

In this procedure, $x_t$ is the input at time step $t$, $h$ is the hidden state of the RNN, $Q$ is an orthogonal matrix that evolves to track an orthonormal basis in tangent space, $J = \frac{df}{dh}$ is the Jacobian of the RNN dynamics with respect to the hidden state, $R$ is the upper-triangular matrix from the QR decomposition, and $\gamma_i$ accumulates the log-magnitudes of the diagonal entries $R_{ii}$. The accumulation begins only after the `warmup` steps, and the final Lyapunov exponent $\lambda_i$ is computed by normalizing $\gamma_i$ by the number of post-`warmup` accumulation steps, which is $K = T - \texttt{warmup}$.

---

**Algorithm 1:** Lyapunov Exponents Calculation

---
**1** Initialize $h, Q$;
**2 for** $t = 1$ **to** $T$ **do**
**3**      $h \leftarrow f(h, x_t)$;
**4**      **if** $t > warmup$ **then**
**5**          $J \leftarrow \frac{df}{dh}$;
**6**          $Q \leftarrow J \cdot Q$;
**7**          $Q, R \leftarrow \texttt{qr}(Q)$;
**8**          $\gamma_i \mathrel{+}= \log(R_{ii})$;
**9** $\lambda_i = \gamma_i / (T - \texttt{warmup})$

---

# D  Lack of transition to chaos for $\alpha = 0.5$

A shown in Fig. 4, networks with $\alpha = 0.5$ do not seem to transition to chaos. For small values of $g$, the MLE increases with $g$ as expected from the stability analysis. Moreover, similarly to other heavy-tailed networks, they hover close to the edge of chaos for a wide range of values of $g$. However, for larger values of $g$ the MLE starts decreasing with $g$ again and, as a result, usually stays negative for all values of $g$. This effect seems to persist for noisy inputs (Fig. 5) and other changes in the parameters of the simulations (Figs. 9 and 10). More work is required to explain the source of this interesting phenomenon. However, since in this study we focus our attention on transition to chaos, for clarity we exclude $\alpha < 1$ from most figures.

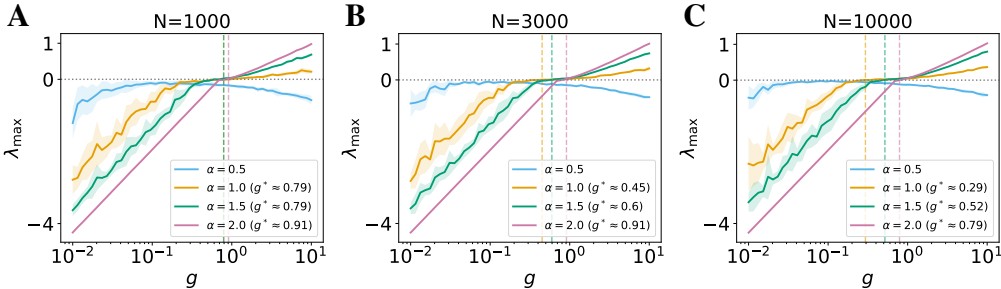

Figure 4: Same as Fig. 2, but with an addition of $\alpha = 0.5$.

# E    Additional results under small noisy input

We replicate our main results under a small i.i.d. Gaussian noise drive (variance = 0.01) sampled at each time step to test the robustness of the quiescent-to-chaotic transition and attractor geometry in more biologically realistic, stimulus-driven settings. Despite the added input variability, which quenches chaos as expected Molgedey et al. [1992], the trends largely mirror the autonomous case.

Figure 5 shows the maximum Lyapunov exponent (MLE) as a function of gain $g$ across network sizes ($N = 1000, 3000, 10000$) and tail indices $\alpha$. Heavier-tailed networks ($\alpha < 2$) exhibit a more gradual increase in MLE and an extended edge-of-chaos regime, consistent with Fig. 2. The transition point shifts leftward with increasing $N$, in line with our mathematical finite-size predictions.

Figure 6 characterizes the attractor dimensionality using Lyapunov spectra, Lyapunov dimension ($D_{\mathrm{KY}}$), and participation ratio (PR). While $D_{\mathrm{KY}}$ and the spectrum remain consistent with the autonomous case, PR displays a U-shaped profile (Fig. 6C), unlike the monotonic rise seen in Fig. 3B. This dip likely reflects a shift in the dominant dynamics: at low $g$, noise drives weak, independent fluctuations across neurons; near the transition, recurrent dynamics compress activity into an elongated low-dimensional manifold; at higher $g$, chaotic expansion increases PR. Thus, while the robustness-dimensionality tradeoff holds under noisy input, noise modulates how variance is distributed across neural modes.

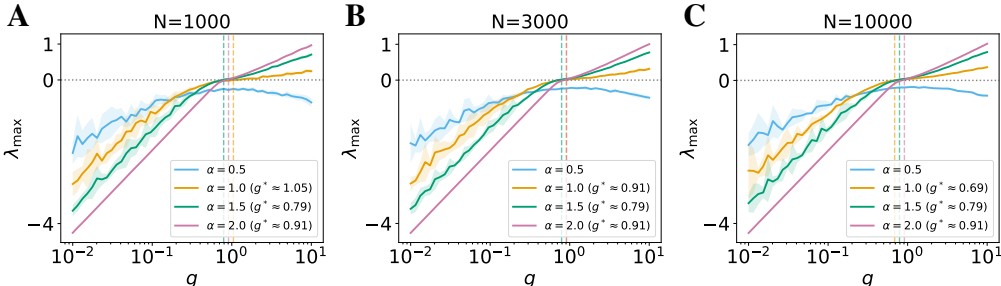

Figure 5: **Effect of network size under small i.i.d. noisy input.** Maximum Lyapunov exponent ($\lambda_{\mathrm{max}}$) as a function of gain $g$ in noisy stimulus-driven recurrent networks with Lévy $\alpha$-stable weight distributions. Curves show mean across 10 trials; shaded regions denote $\pm 1$ SD. Each panel corresponds to a different network size: (A) $N = 1000$, (B) $N = 3000$, and (C) $N = 10000$. Curves show mean across 3 trials; shaded regions denote $\pm 1$ SD. As in the autonomous case, if a transition exists, then heavier-tailed networks exhibit a slower transition and wider critical regime near $\lambda_{\mathrm{max}} = 0$. The critical gain $g^*$ (dashed line) shifts leftward with increasing $N$, consistent with finite-size theory.

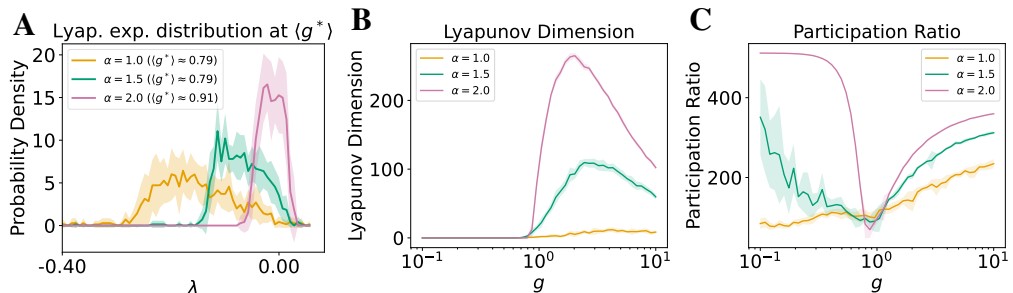

Figure 6: **Attractor geometry under noisy input (N = 1000).** Curves show mean across 10 trials; shaded regions denote $\pm 1$ SD. (A) Lyapunov exponent distributions at the estimated transition point $g^*$. Heavier-tailed networks exhibit fewer near-zero exponents, indicating a compressed slow manifold. x-axis truncated at the left to omit near-zero tails for clarity. (B) Lyapunov dimension declines with heavier tails, confirming lower attractor dimensionality as found in the autonomous networks. (C) Participation ratio shows a distinct dip near transition to chaos before rising, unlike the monotonic profile observed in the autonomous case, but it is consistently lower in heavier-tailed networks otherwise.

# F   Visualizations of multiple realizations of Lyapunov spectrum

To assess the variability across realizations of network connectivity, we visualize the full Lyapunov spectrum from three independent trials (different seeds) for networks with $N = 1000$, across both autonomous and noisy stimulus-driven settings. These spectra are computed near the estimated critical gain $g^*$ (obtained in Figs. 2 and 5), where the maximum Lyapunov exponent $\lambda_{\max}$ first crosses zero in each condition. The same value of $\langle g^* \rangle$, computed by averaging $g^*$ of multiple runs, is used across different seeds in these figures. The actual transition point can vary in each realization. Moreover, due to the finite resolution of the grid of $g$ values used in simulations, $g^*$ is overestimated in each seed. Thus, in some realizations, the right edge of the histogram may exceed $0$, and the average histograms presented in Figs. 3A and 6A can feature some positive Lyapunov exponents. A more precise estimate could be obtained through a finer-grained or binary search over gain values near the transition point. However, this additional numerical precision would unlikely affect our overall conclusions.

In both autonomous (Fig. 7) and noisy stimulus-driven cases (Fig. 8), Gaussian networks exhibit a dense cluster of exponents near zero, indicative of a broad slow manifold. In contrast, heavier-tailed networks (lower $\alpha$) show more widely dispersed exponents with fewer near-zero values, consistent with the compression of the slow manifold described in Section 4.3.1. Despite random initialization, the qualitative trend—greater spectrum spread and fewer marginal directions as $\alpha$ decreases—remains consistent across seeds. Notably, in the noisy stimulus-driven case (Fig. 8), the exponents tend to shift downward, and their distributions become more skewed, particularly for heavier-tailed networks. These effects likely reflect interactions between stochastic input and the network's intrinsic dynamics, where the noise quenches the chaos.

Together, these visualizations reinforce our claim that heavy-tailed connectivity leads to systematically lower-dimensional attractors, regardless of input conditions or initializations.

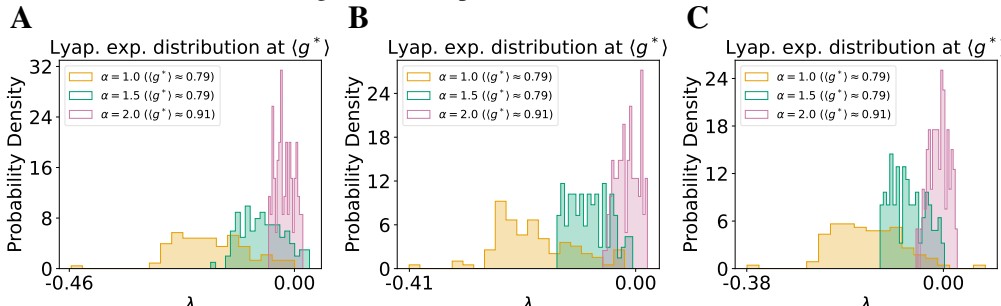

Figure 7: **Full Lyapunov spectra across random initializations in autonomous networks ($\mathbf{N = 1000}$).** Each panel shows the Lyapunov exponent distributions near estimated $g^*$ for an independent seed. Across seeds, heavier-tailed networks (lower $\alpha$) exhibit a broader spectrum with fewer exponents near zero, indicating reduced slow-manifold dimensionality compared to Gaussian networks.

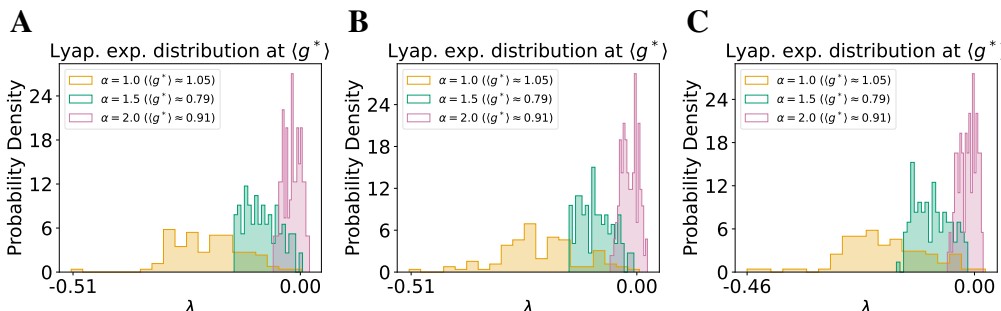

Figure 8: **Full Lyapunov spectra across random initializations in noisy stimulus-driven networks ($\mathbf{N = 1000}$).** Each panel shows the Lyapunov exponent distributions at the estimated critical gain $g^*$ for an independent seed. Spectra under noise remain qualitatively similar to the autonomous case.

# G Robustness of results shown in Fig. 2

Since we only examine the maximum (top-1) Lyapunov exponent in Fig. 2, the number of top exponents computed (denoted `k_LE` in the codebase) is irrelevant as long as `k_LE` > 1. Throughout Fig. 2 and this appendix, we use the default `k_LE` = 100. Additionally, we exclude an initial warmup period before accumulating exponents to avoid contamination from transients (Appendix C). In Fig. 2, the network is run for $T = 3000$ steps, and Lyapunov exponents are accumulated over the final $K = 100$ steps.

Note that computational cost increases with network size $N$, number of exponents `k_LE`, accumulation duration $K$, and total time steps $T$. Here, we verify that our results in Figs. 2 and 5 are robust to these choices by comparing the default configuration against two more computationally demanding variants, keeping all else fixed:

1. Accumulating exponents over a longer period ($K = T - \texttt{warmup} = 150$);
2. Running the network for longer total time ($T = 4000$ with `warmup` of 3900, fixing $K = 100$).

The results are shown in Fig. 9 (autonomous) and Fig. 10 (noisy). All curves remain nearly identical across conditions, demonstrating that our findings are not sensitive to the specific accumulation duration or simulation length. In practice, using $T = 3000$ and $K = 100$ strikes a good balance between computational efficiency and accuracy, especially for large $N$. These findings validate that the trends reported in Figs. 2 and 5 are robust, and additional compute is not necessary. Note that the effect of network size $N$ has been evaluated in Figs. 2 and 5, in which the critical transition $g^*$ shifts to the left as $N$ increases due to the finite-size effect.

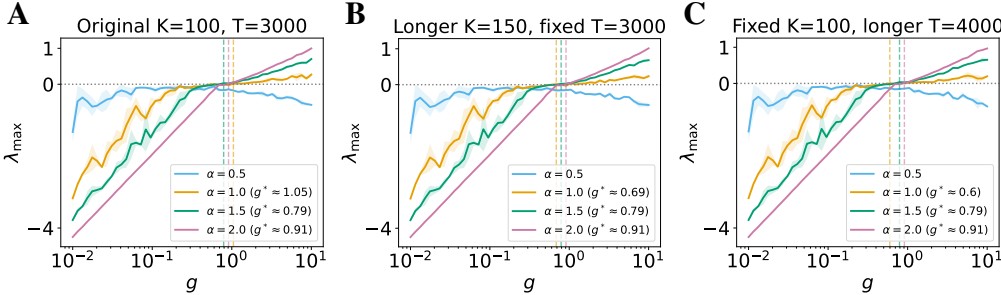

Figure 9: **Robustness of MLE to time horizon and accumulation duration (autonomous, N = 1000).** Curves show mean across 3 trials; shaded regions denote ±1 SD. (A) Default configuration: $T = 3000$, $K = 100$; (B) Longer accumulation: $K = 150$; (C) Longer sequence: $T = 4000$ with $K = 100$. Results are nearly identical, confirming that the choice of $T$ and $K$ does not affect the reported trends.

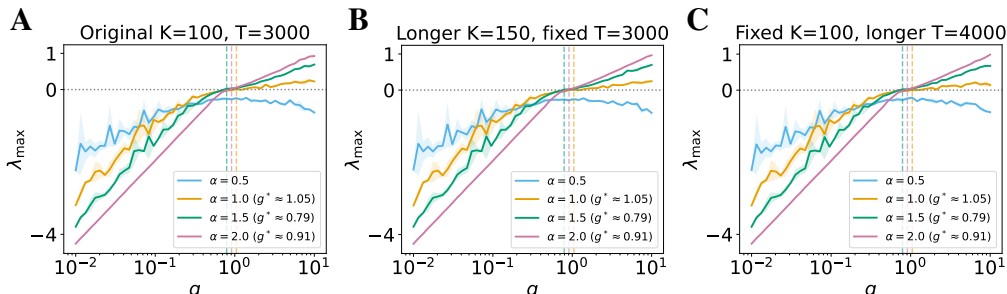

Figure 10: **Robustness of MLE to time horizon and accumulation duration (noisy stimulus-driven, N = 1000).** Curves show mean across 3 trials; shaded regions denote ±1 SD. (A) Default configuration: $T = 3000$, $K = 100$; (B) Longer accumulation: $K = 150$; (C) Longer sequence: $T = 4000$ with $K = 100$. Results remain stable, indicating that stochastic input does not impact the robustness of MLE computation.

# H   Robustness of results shown in Fig. 3A

We showed the representative top 100 Lyapunov exponents (`k_LE` = 100) in Fig. 3A using networks of size $N = 1000$. As we are primarily interested in the region near $\lambda = 0$, this choice of `k_LE` is sufficient and larger values do not change the results.

In both Fig. 3A and the visualizations in Appendix F, networks were evolved for $T = 3000$ time steps, with the exponents accumulated over the final $K = 100$ steps, after an initial warmup. As with all our experiments, computation becomes more expensive as the network size $N$, number of exponents `k_LE`, accumulation duration $K$, and total time steps $T$ increase. Here we test the robustness of our findings in Fig. 3A by varying these computational parameters, holding all else fixed:

1. Increasing network size to $N = 3000$ (panels A);
2. Accumulating over a longer time window $K = 150$ (panels B);
3. Increasing the total simulation length to $T = 4000$ while maintaining $K = 100$ (using a longer warmup of 3900, panels C).

The resulting spectra, shown below in both autonomous (Fig. 11) and noisy stimulus-driven networks (Fig. 12), are qualitatively the same as the original results. The shape of the Lyapunov spectrum remains consistent: Gaussian networks show a dense band near zero, and heavier-tailed networks exhibit broader spectra with fewer exponents near zero. These results confirm that our main finding—compression of the slow manifold with decreasing $\alpha$—is robust across a range of network sizes and simulation settings. For large-scale experiments, using $N = 1000$, $T = 3000$, and $K = 100$ provides a reliable and computationally efficient default.

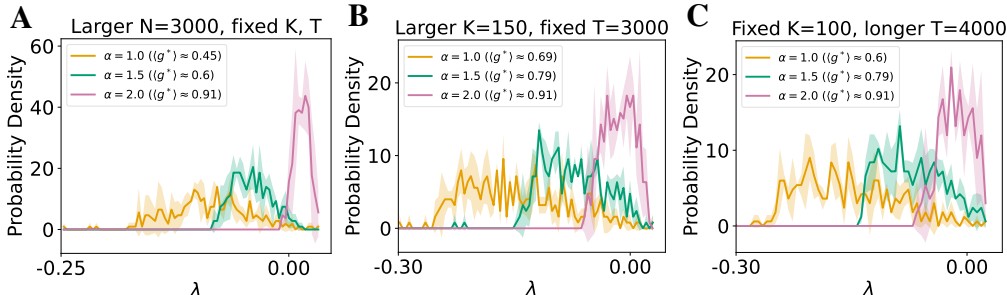

Figure 11: **Robustness of Lyapunov spectra to simulation and accumulation parameters (autonomous).** Curves show mean across 3 trials; shaded regions denote $\pm 1$ SD. Mean Lyapunov spectra near $g^*$ under three conditions: (A) larger network size ($N = 3000$); (B) longer accumulation period ($K = 150$); (C) longer total simulation length ($T = 4000$). Both (B) and (C) use $N = 1000$. The compressed spectrum in heavier-tailed networks remains consistent across all conditions.

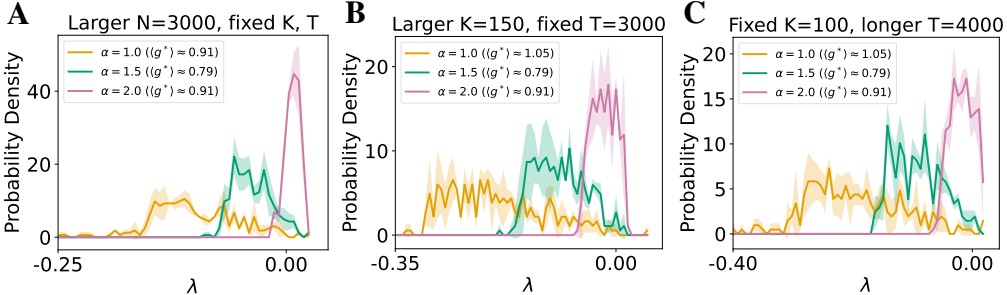

Figure 12: **Robustness of Lyapunov spectra to simulation and accumulation parameters (noisy input).** Same conditions as Fig. 11, but for noisy stimulus-driven networks. Despite stochastic input, heavier-tailed networks continue to exhibit a wider Lyapunov spectrum with fewer marginally stable directions as indicated by the Lyapunov exponents being zero.

# I Robustness of results shown in Fig. 3B,C

**Lyapunov dimension** We use the full Lyapunov spectrum to compute the results shown in Fig. 3B due to the definition of Lyapunov dimension (Eqn. 4), hence $\texttt{k\_LE} = N = 1000$ in Fig. 3B. We simulate the dynamics over a total number of $T = 2950$ steps, and use the last $K = 50$ steps to compute the Lyapunov dimension.

**Participation ratio** To ensure a well-defined participation ratio (PR, Eqn. 5), we require $K > N$, where $N$ is the network size and $K = T - \texttt{warmup}$ denotes the number of time steps used for computing PR after the network has evolved for a number of $\texttt{warmup}$ steps. This condition guarantees that the empirical covariance matrix $S$, computed from $K$ samples of $N$-dimensional hidden states, is full-rank and not rank-deficient. When $K \leq N$, $S$ becomes singular or ill-conditioned, leading to unreliable estimates of its eigenvalue spectrum and thus of the participation ratio. In Fig. 3C, we use $T = 2900 + N + 50 = 3950$, meaning 2900 $\texttt{warmup}$ steps with an accumulation period over the last 1050 steps.

Note that the computation cost increases as $N$, $\texttt{k\_LE}$, $K$, and $T$ increase.

Here we show our results in Fig. 2 is robust, meaning it is consistent with the more computationally demanding case(s) with all else fixed:

1. Bigger network size $N = 3000$ (panels A);

2. Longer accumulation period $K = 100$ for computing Lyapunov dimension, and longer $K = 1100$ for computing participation ratio (panels B);

3. Longer time trajectory $T = 3950$ for computing Lyapunov dimension and $T = 4950$ for computing participation ratio (panels C).

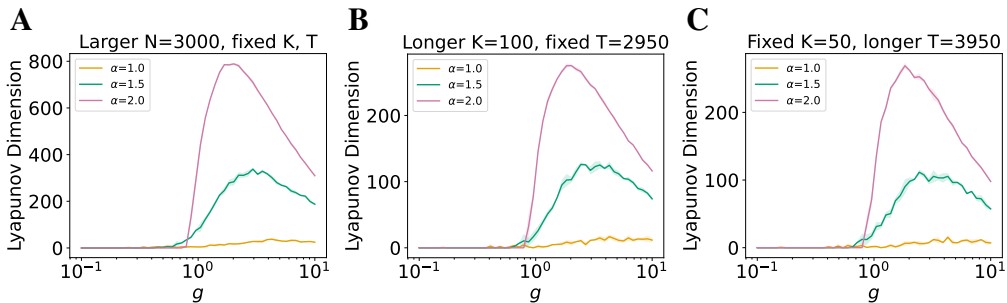

Figure 13: **Robustness of Lyapunov dimension to simulation parameters (autonomous).** (A) Larger network size $N = 3000$; (B) Longer accumulation period $K = 100$; (C) Longer total sequence $T = 3950$ with $K = 50$. All trends remain consistent with those in Fig. 3B.

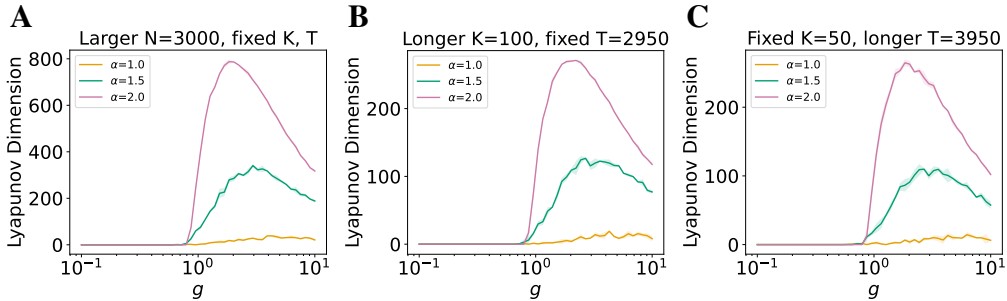

Figure 14: **Robustness of Lyapunov dimension to simulation parameters (noisy).** Same settings as Fig. 13, but with i.i.d. Gaussian input. Results are stable across conditions, confirming robustness of $D_{\mathrm{KY}}$ in noisy networks, consistent with those in Fig. 6B.

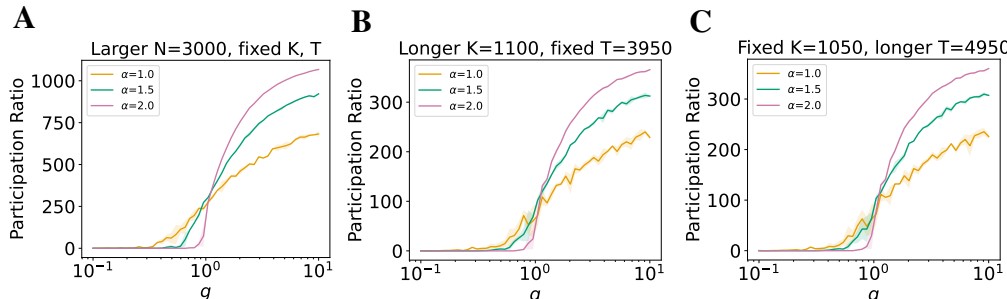

Figure 15: **Robustness of participation ratio to simulation parameters (autonomous).** (A) Larger network size $N = 3000$, $K = 3050$; (B) Longer accumulation period $K = 1100$; (C) Longer sequence $T = 4950$, $K = 1050$. All curves are consistent with Fig. 3C, confirming stability of PR under varying conditions.

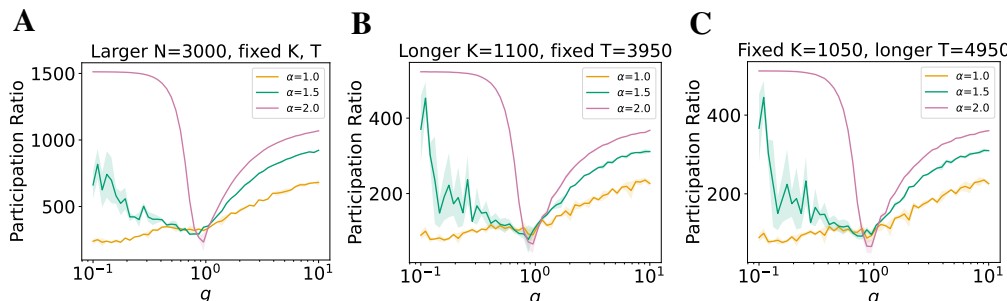

Figure 16: **Robustness of participation ratio to simulation parameters (noisy).** Same configurations as Fig. 15, but under i.i.d. Gaussian input. The non-monotonic profile and overall trends in PR are preserved across all tested conditions, consistent with those in Fig. 6C.

## J    Behavior of the quenched transition point as a function of $N$

In finite-sized quenched networks, the location of the transition point fluctuates between realizations of the weight matrix. Since our annealed theory does not offer any insight into the nature of these fluctuations, we resorted to numerical simulations to study how the statistics of $g^*$ scale with $N$. The results for the representative case of $\alpha = 1$ are shown in Fig. 17. The mean location of the transition point scales like $1/\ln N$, in line with our theoretical predictions (Fig. 17A). The annealed prediction seems to underestimate the true mean over quenched realizations. The standard deviation of $g^*$ decreases with $N$ at a comparable rate as the mean (Fig. 17B). The coefficient of variation of $g^*$ falls off slowly in the studied range of $N$ (Fig. 17C), suggesting that the location of the transition may be (weakly) self-averaging [Wiseman and Domany, 1998] in this system.

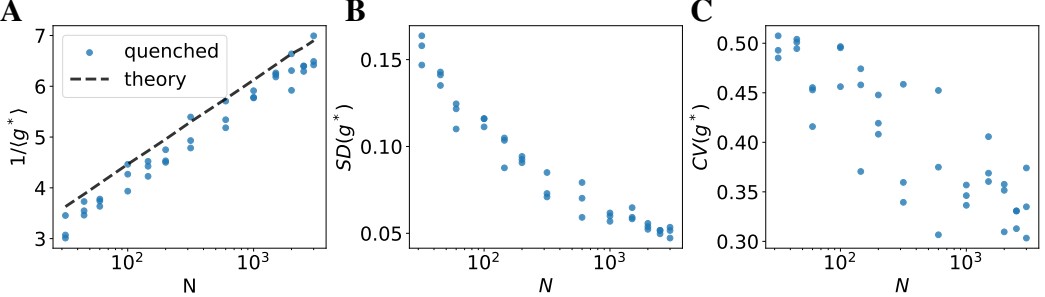

Figure 17: **Statistics of $g^*$ in quenched networks as functions of $N$.** Note the logarithmic scale on the x-axis. Each point corresponds to the statistics estimated using 100 independent realizations of the weight matrix. For each value of $N$, we included three data points that correspond to independent estimates calculated based on different random seeds. (A) Reciprocal of the mean. (B) Standard deviation. (C) Coefficient of variation.

## K The effect of mega-synapses on dynamics

To test whether robustness and low dimensionality arise from global heavy-tailed statistics or a few extreme "mega-synapses," we pruned recurrent weights in a network of size $N = 1000$ by absolute magnitude (bottom 95%, top 1%, top 3%) and report the results averaged across three trials. The slow transition vanished when top outliers were removed, shifting the critical gain $g^*$ to larger values, whereas pruning the weakest 95% had little effect (Fig. 18).

Similarly, removing the bottom percentage of weights has very little effect on the general trend of attractor dimensionality. However, when top outliers are removed, the changes are more nuanced: the attractor dimensionality for heavy-tailed weights increases in the chaotic regime, while the transition to chaos is pushed to a larger $g^*$ when more top outlier weights are pruned as mentioned above (Figs. 19, 20). The general ranking of dimensionality by $\alpha$ is largely consistent with the main paper for both dimension measures, though the max dimensionality of $\alpha = 1.5$ is comparable to that of $\alpha = 2$ over a range of $g$ when top outliers are pruned.

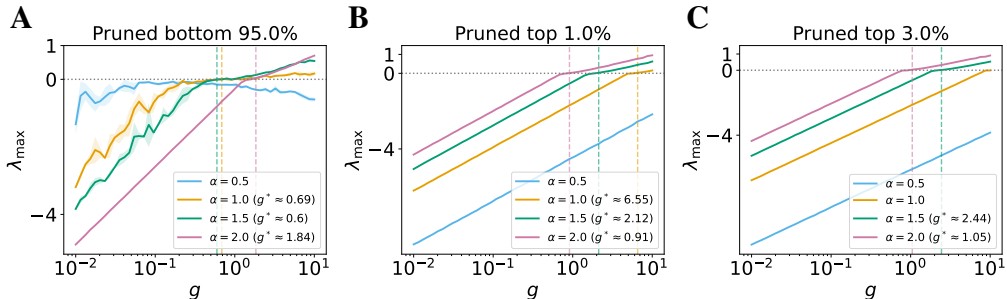

Figure 18: **Effect of pruning on critical gain $g^*$.** (A) bottom 95% removed. (B) top 1% removed. (C) top 3% removed.

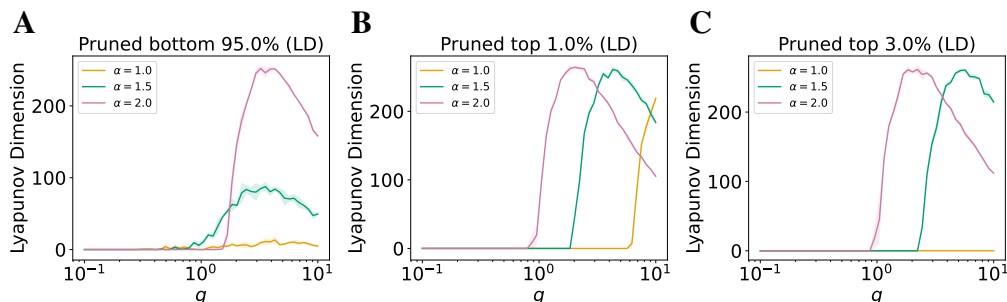

Figure 19: **Effect of pruning on Lyapunov dimension.** (A) bottom 95% removed. (B) top 1% removed. (C) top 3% (LD) removed.

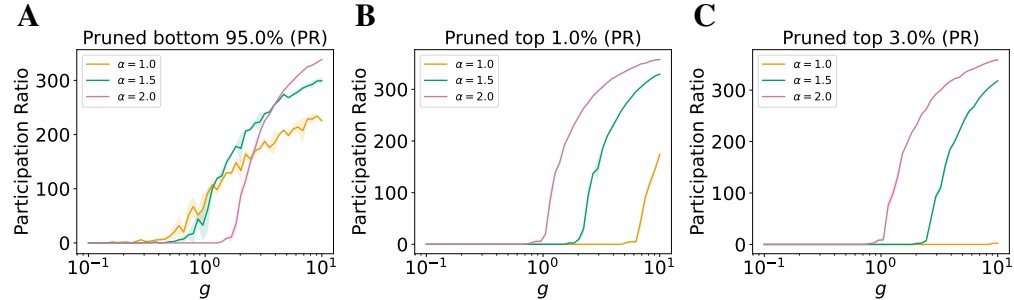

Figure 20: **Effect of pruning on participation ratio (PR).** (A) bottom 95% removed. (B) top 1% removed. (C) top 3% removed.

## L  Information processing in heavy-tailed recurrent neural networks

To examine whether our results extend to structured external inputs (and toward learned settings), we provide a proof-of-concept through a reservoir-computing experiment on the delayed-memory XOR task (a similar task is used in [Huh and Sejnowski, 2018]). We use networks of size $N = 1000$ and report average performance across three trials. Specifically, in the XOR task, each trial presents two binary stimulus vectors $s_1, s_2$ separated by silent delays; after the second delay, the readout must report $\mathrm{XOR}(s_1, s_2)$, requiring short-term maintenance of both stimuli and a nonlinear decision rule.

Across gains $g$, heavy-tailed reservoirs exhibited a broader and more stable operating regime than Gaussian reservoirs: the transition to chaos was slower and more robust (Fig. 21A), and task performance remained high over a wider range of $g$ (Fig. 21B). These observations suggest that the extended critical regime of heavy-tailed networks can enhance robustness and performance without fine-tuning, with potential benefits for machine learning applications.

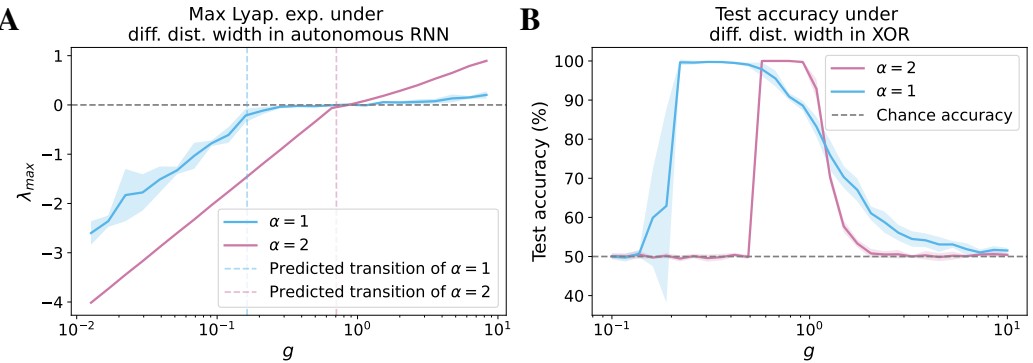

Figure 21: **Delayed-memory XOR with heavy-tailed reservoirs. (A)** Dynamics across gain $g$ as measured by maximum Lyapunov exponent for Gaussian (pink) vs. heavy-tailed reservoirs (blue), showing a slower, more robust transition to chaos in the latter. **(B)** Task accuracy of a linear readout on the same reservoirs, with heavy-tailed networks maintaining strong performance over a broader range of $g$.

# M  Additional details

## M.1  Experiments compute resources

All experiments reported in this paper can be reproduced using CPUs only; no GPUs are required. The only exception is Fig. 1, for which we strongly recommend using a single GPU to avoid potential compatibility issues with the JAX package. Jobs were executed on a compute cluster using a maximum of 4 CPU cores and 20 GB of memory per task (which is a very conservative allocation; for networks of size $N = 1000$, for example, 5 GB is likely sufficient). Each experimental run was allocated up to 24 hours of wall-clock time. Most runs completed well within this limit, with small networks $N = 1000$ usually completed within 5 hours running serially over a grid of 50 gain $g$ values, three tail indices $\alpha$, and over 3 trials. Storage requirements were modest and standard across all runs. While additional preliminary experiments were conducted during development, they did not require significantly more compute and are not reported in the final results.

## M.2  Licenses for existing assets

This project makes use of several open-source Python packages. While the main paper does not formally cite each package, we acknowledge their use here and ensure full transparency by providing all code and dependencies in the released repository. Below we list each core package, its version, license, and citation if applicable:

| Package | Version | License | URL | Citation |
|---------|---------|---------|-----|----------|
| `jax, jaxlib` | v0.4.38 | Apache 2.0 | `https://github.com/google/jax` | [Bradbury et al., 2018] |
| `numpy` | v1.26.4 | Modified BSD | `https://numpy.org/` | [Harris et al., 2020] |
| `scipy` | v1.15.2 | BSD | `https://scipy.org/` | [Virtanen et al., 2020] |
| `torch` | v2.7.0 | Modified BSD | `https://pytorch.org/` | [Paszke et al., 2019] |
| `tensorflow, keras` | v2.19.0, v3.9.2 | Apache 2.0 | `https://www.tensorflow.org/` | [Martín et al., 2015] |
| `matplotlib` | v3.10.1 | PSF | `https://matplotlib.org/` | [Hunter, 2007] |
| `tqdm` | v4.67.1 | MIT | `https://github.com/tqdm/tqdm` | – |

Table 1: Third-party Python packages used in this paper, with version numbers, licenses, source URLs, and citations where applicable.

Python versions `>=3.10` and `<3.13` are recommended. All software dependencies are installable via `pip` using the provided `requirements.txt`. No proprietary assets were used in this study.

