# OpenReview forum: "Slow Transition to Low-Dimensional Chaos in Heavy-Tailed Recurrent Neural Networks"
_NeurIPS.cc/2025/Conference — NeurIPS 2025 poster_

### Official Review · Reviewer_V5Z3 · 2025-06-17

**Clarity:** 3
**Significance:** 3
**Originality:** 3
**Rating:** 5
**Confidence:** 4

**Summary:**

This paper analyzes the edge-of-chaos dynamics of the recurrent neural networks (RNN) with heavy-tailed connections. The authors show that comparing to the RNN using Gaussian connections, the heavy-tailed RNN exhibit a broader gain regime near the edge of chaos, while has few dimensionality of the attractor. This result reveals a trade-off of the robustness and the representation ability of the edge-of-chaos state in the RNN.

**Questions:**

1. Can the main results be recovered when the activation function $\phi$ is unbounded (ReLU, for example)?
2. Could the authors provide an intuitive explanation for why, during the mathematical analysis, the weight matrix $W_{ij}$  can be treated as being redrawn at each time step?

**Ethical Concerns:**

["NO or VERY MINOR ethics concerns only"]

**Final Justification:**

The rebuttal addresses most of my concerns, and I maintain my postive score.

**Limitations:**

Please see the weaknesses.

**Paper Formatting Concerns:**

There is no paper formatting concerns.

**Quality:**

3

**Strengths And Weaknesses:**

### Strengths

1. The paper is clearly written and easy to follow.
2. The main result that the robustness and dimensionality are trade-off is interesting and may be important for deeper understanding of the nonlinear neural dynamics in the brain.
3. The authors present a systematic set of experiments in both the main paper and the supplementary materials, making their experimental results highly convincing.

### Weaknesses

1. It is not clear whether the results can be extended to the cases that the weights of the RNN are learned instead of randomly sampling, or when the external inputs are structured, limited the impact of the paper.
2. The variation of the dimensionality and the participation ratio near the edge of chaos is not well explained theoretically. It is better to give more explanation (some intuitive explanations are also helpful).

---

> ### Author Rebuttal · Authors · 2025-07-29
>
> We deeply appreciate the reviewer’s comprehensive feedback. We sincerely thank the reviewer for acknowledging the clarity and soundness of our findings, and pointing out their potential significance to a deeper understanding of the nonlinear neural dynamics in the brain.
>
> The reviewer’s main concerns are regarding how the results extend to trained RNNs that are task-driven, in addition to seeking clarifications on our analysis. Here, we address each by providing further explanations and intuitions, and state the preliminary result on the robustness of heavy-tailed weights benefits reservoir computing performance in a temporal XOR task (as a proof-of-concept for the robustness benefits in information processing). **Together with the discussion below, does this sufficiently address the reviewer’s concerns?**
>
> > It is not clear whether the results can be extended to the cases that the weights of the RNN are learned instead of randomly sampling, or when the external inputs are structured, limited the impact of the paper.
>
> Thank you for this insightful comment. We agree that assessing heavy‑tailed dynamics under learned weights and structured inputs is an exciting next frontier. Because training introduces many additional factors, such as optimizer noise, curriculum effects, and loss‑surface geometry, to name a few, we believe it would be best suited for a dedicated follow‑up study that we hope to pursue. We also warmly welcome community efforts in that direction.
>
> To offer an initial glimpse and better situate our results in information processing, we will add new reservoir-computing experiments on the delayed memory XOR task as a proof-of-concept for the case of structured external inputs (as suggested by the reviews).
>
> **Task:** In this setting, each trial presents two binary stimulus vectors $s_1, s_2$, each drawn from one of two classes and separated by silent delay periods; after the second delay, the readout must report $\operatorname{XOR}(s_1, s_2)$. This requires the network to store both stimuli in short‑term memory and apply a nonlinear decision rule. A similar delayed‑XOR task is studied in Huh & Sejnowski (NeurIPS 2018).
>
> We observe that heavy-tailed networks exhibit a broader gain regime with strong performance compared to Gaussian networks, aligning with our main results showing heavier-tailed networks exhibit a slower, more robust transition to chaos, which can translate to task benefits. This would provide a concrete proof-of-concept that the extended critical regime can enhance robustness and performance without fine-tuning, which is beneficial in ML applications. The revised manuscript, if accepted, will report this result.
>
> > The variation of the dimensionality and the participation ratio near the edge of chaos is not well explained theoretically. It is better to give more explanation (some intuitive explanations are also helpful).
>
> Thanks for the constructive feedback. We believe that the qualitative intuition behind the large disparity in Lyapunov dimensions between Gaussian and heavy-tailed networks can be obtained through Fig. 3A: the Lyapunov exponents are more dispersed for networks with heavier tails. Thus, in heavy-tailed networks, only a small number of leading Lyapunov exponents become positive when being close to the edge of chaos, leading to lower values of Lyapunov dimension. This, in turn, is driven by the more heterogeneous distribution of eigenvalues of the weight matrix in heavy-tailed networks. However, the exact correspondence between the eigenvalues of the weight matrix and the Jacobian is nontrivial. Consequently, we were not able to obtain analytical predictions of the Lyapunov and participation ratio dimensions, but we will add this intuition to Section 4.3.
>
> > Can the main results be recovered when the activation function is unbounded (ReLU, for example)?
>
> If the activation function is unbounded, there is no longer guarantee that the dynamics does not diverge. Therefore, while the existence of the transition can still be assured for a large class of activation functions (see our response to Reviewer ga7q), the nature of the dynamics above the transition may qualitatively change depending on the activation function.
>
> > Could the authors provide an intuitive explanation for why, during the mathematical analysis, the weight matrix W_{ij} can be treated as being redrawn at each time step?
>
> In the case of feedforward networks, weights at different layers are usually drawn independently at initialization, so our analysis is exact in this case. In the case of RNNs, however, the weights usually stay constant across time steps, so this analysis is an approximation. A priori, there is no obvious reason why this approximation should provide good estimates of the quantities of interest, but in some cases it can become exact in the limit of infinite system size (e.g., in sparsely diluted, asymmetric version of the Hopfield network, see Derrida, Gardner, & Zippelius, A., EPL, 1987). The basic intuition is that the correlations induced by the quenched disorder decrease with the size of the network and become negligible in infinite networks.
>
> Although we do not currently know whether the annealed approximation becomes exact in the infinite RNNs with heavy-tailed weights, our numerical experiments indicate that the relative mean squared error (i.e., the MSE between annealed and quenched transition $g$ divided by the squared mean annealed transition $g$) of the estimated location of the transition does significantly decrease with $N$, at least in the range between $N=32$ and $N=3000$.

---

> > ### Comment · Reviewer_V5Z3 · 2025-08-04
> > **Thank you for the rebuttal**
> >
> > The authors’ rebuttal addresses most of my concerns. However, several key issues remain:
> >
> > 1. The new XOR experiments show that the heavy-tailed network achieves better performance compared to the Gaussian baseline, suggesting improved robustness. However, this does not explicitly demonstrate the trade-off between robustness and dimensionality. Could the authors provide further analysis to clarify this relationship in the context of their task?
> >
> > 2. Since unbounded activations are more widely used in deep learning, expanding the discussion on their implications would further strengthen the paper’s relevance. Such an analysis could offer deeper insights into training deep networks, thereby increasing the significance of the work.

---

> > > ### Author Response · Authors · 2025-08-07
> > >
> > > We are glad to hear that most of the reviewer's concerns have been addressed. We deeply appreciate their thoughtful feedback and engagement in the rebuttal process.
> > >
> > > > The new XOR experiments show that the heavy-tailed network achieves better performance compared to the Gaussian baseline, suggesting improved robustness. However, this does not explicitly demonstrate the trade-off between robustness and dimensionality. Could the authors provide further analysis to clarify this relationship in the context of their task?
> > >
> > > We thank the reviewer for their insightful feedback. We agree that while the XOR task suggests improved robustness, a more extensive analysis would be needed to demonstrate the tradeoff in action. However, in our opinion, such an analysis would likely involve a broader suite of tasks to be thorough. The relevant results, as well as their interpretations and discussion, would significantly exceed the page limit and would not be appropriate to relegate to the Appendix.
> > >
> > > The paper’s primary goal is to isolate untrained network statistics and show qualitative dynamical signatures of heavy-tailed weight networks. While we agree that a full robustness-dimensionality trade-off analysis is important and impactful, we believe pursuing it here would shift the narrative and dilute the main contribution. We instead view this as an exciting direction for future work. We hope this submission will provide the foundational work to open a new avenue and inspire the community to pursue (see lines 363-372).
> > >
> > > > Since unbounded activations are more widely used in deep learning, expanding the discussion on their implications would further strengthen the paper’s relevance. Such an analysis could offer deeper insights into training deep networks, thereby increasing the significance of the work.
> > >
> > > We agree with and appreciate the reviewer’s constructive insight. If accepted, we will add to the discussion in the revision, with appropriate rewording based on the following:
> > >
> > > Our analysis is based on the linear stability analysis of the quiescent fixed point, which exists for activation functions that satisfy $\phi(0)=0$. Moreover, we utilize the linearization, which is valid as long as the activation function has a linear part around the origin, i.e., it admits an expansion of the form $\phi(x) = ax + o(x)$. The existence of the transition is then assured by the linear stability analysis.
> > >
> > > The analysis can be extended to cases such that $\phi(0) \neq 0$, by expanding the update equation around the resulting nonquiescent fixed point. The analysis is complicated by the fact that now the location of the stable fixed point itself may change with the order parameter $g$, but the qualitative nature of the transition should generically remain unchanged.
> > >
> > > Although the existence of a finite-size transition uncovered in our work is very general, the nature of the dynamics above the transition point (i.e., after the fixed point becomes unstable) depends on further details of the activation function. For example, if the activation function is linear, the activity diverges once the fixed point becomes unstable. Similarly, ReLU should lead to the divergence for large $g$. In general, a sublinear activation function is needed to control the divergence. We expect our results to hold for any smooth saturating nonlinearity with features described above.

---

### Official Review · Reviewer_ga7q · 2025-06-30

**Clarity:** 3
**Significance:** 3
**Originality:** 3
**Rating:** 5
**Confidence:** 4

**Summary:**

This paper investigates recurrent neural networks (RNNs) initialized with heavy-tailed distributions and examines their dynamic properties. Using metrics such as transition points, Lyapunov exponents, Lyapunov dimension, and participation ratio, the authors demonstrate that heavier-tailed synaptic connectivity expands the regime of stable, edge-of-chaos dynamics, while simultaneously reducing the dimensionality of the resulting activity.

**Questions:**

Please see 'Weakness'.

**Ethical Concerns:**

["NO or VERY MINOR ethics concerns only"]

**Final Justification:**

This is a well-written paper that addresses a gap on heavy-tailed RNNs. The authors have addressed all my concerns.

**Limitations:**

yes

**Paper Formatting Concerns:**

No.

**Quality:**

3

**Strengths And Weaknesses:**

This paper provides detailed mathematical proofs and visualizations, along with a comprehensive set of simulations.

My main concern is that all the networks studied are untrained and rely solely on initialization (as the authors acknowledge). How do the properties of heavy-tailed recurrent neural networks evolve during training? Furthermore, do the weight distributions remain heavy-tailed after training?

Another point of concern is the use of activation function. What role does the choice of activation function play in the observed dynamics? Would alternative nonlinearities, such as ReLU, significantly alter the findings?

Although the model draws inspiration from biological systems, the study lacks a direct comparison with neural data.

Given that the authors acknowledge these limitations and have nonetheless conducted a thorough analysis of initialization-only networks, my score will be borderline but positive.

---

> ### Author Rebuttal · Authors · 2025-07-29
>
> We deeply appreciate the reviewer’s comprehensive feedback. We sincerely thank the reviewer for acknowledging the thoroughness of our analysis on fixed heavy-tailed networks, and that their main concerns regarding limitations are captured in our discussion section of the submission.
>
> The reviewer’s main concerns are regarding further results beyond the fixed heavy-tailed networks, e.g., analysis of trained networks, as well as the role of activation functions and comparison to neural data. Here, we address each with our justifications and any related intuition, as well as additional results on reservoir computing networks. **Together with the discussion below, does this sufficiently address the reviewer’s concerns?**
>
> > My main concern is that all the networks studied are untrained and rely solely on initialization (as the authors acknowledge). How do the properties of heavy-tailed recurrent neural networks evolve during training? Furthermore, do the weight distributions remain heavy-tailed after training?
>
> We appreciate this constructive feedback. Our aim is to establish a clean baseline by analyzing untrained heavy‑tailed RNNs, much as what Schoenholz et al. (ICLR, 2017) did for **untrained** Gaussian deep feedforward networks, where they found trainability hinges on initializing near the edge of chaos. How the heavy‑tailed weight distribution evolves under learning is a rich follow‑up problem whose answer depends on the training regime.
> In “lazy’’ or neural‑tangent‑kernel limits (Jacot et al., NeurIPS, 2018; Chizat et al., NeurIPS, 2019), weights move only infinitesimally, so we would expect the heavy tails (and the broadened critical region we report) to persist. By contrast, a more sophisticated investigation is required in “rich” / feature‑learning regimes, such as extending our finite‑size theory to the stochastic dynamics of SGD.
>
> Exploring how our finite‑size theory interacts with learning dynamics is therefore an exciting direction we leave for future work; we will emphasize this point (lines 363-366) in the revision, if accepted, to highlight the open questions and invite community investigation.
>
> Additionally, following the reviews, if accepted, we will include in the appendix our new result on a delayed memory XOR task under a reservoir computing setup (see task description in the rebuttal response to Reviewer V5Z3). In this setting, we observe that heavy-tailed networks exhibit a broader gain regime with strong performance compared to Gaussian networks. This aligns with our main results showing heavier-tailed networks exhibit a slower, more robust transition to chaos, which translates to task benefits. This would provide a concrete proof-of-concept that the extended critical regime can enhance robustness and performance without fine-tuning, which is beneficial in ML applications.
>
> We hope that the reviewer would find this addition a good middle ground between adding an extensive study of training (which, in our opinion, would distract the main narrative and findings of this paper) v.s. the current version of the paper.
>
> > Another point of concern is the use of activation function. What role does the choice of activation function play in the observed dynamics? Would alternative nonlinearities, such as ReLU, significantly alter the findings?
>
> We thank the reviewer for pointing out this important question. Our analysis is based on the linear stability analysis of the quiescent fixed point, which exists for activation functions that satisfy $\phi(0)=0$. Moreover, we utilize the linearization, which is valid as long as the activation function has a linear part around the origin, i.e., it admits an expansion of the form $\phi(x) = ax + o(x)$. The existence of the transition is then assured by the linear stability analysis.
>
> The analysis can be extended to cases such that $\phi(0) \neq 0$, by expanding the update equation around the resulting nonquiescent fixed point. The analysis is complicated by the fact that now the location of the stable fixed point itself may change with the order parameter $g$, but the qualitative nature of the transition should generically remain unchanged.
>
> The nature of the dynamics above the transition point (i.e., after the fixed point becomes unstable) depends on further details of the activation function. For example, if the activation function is linear, the activity diverges once the fixed point becomes unstable. In general, a sublinear activation function is needed to control the divergence. We expect our results to hold for any smooth saturating nonlinearity with features described above. The case of ReLU seems more complex, and we expect it to behave in a more complex manner, leading to divergent dynamics for large $g$.
>
> We plan to add a short discussion of this issue in the revised version of the manuscript.
>
> > Although the model draws inspiration from biological systems, the study lacks a direct comparison with neural data.
>
> Thank you for raising this thoughtful point. Our goal in the present work is to lay a clear foundation by isolating the consequences of heavy‑tailed connectivity, rather than fitting a specific dataset. We do, however, cite empirical studies that report heavy‑tailed synaptic weight distributions being common in the brain across species (lines 19-25), which motivated our study. We agree that this is a valuable and exciting next step to build on our foundation and compare directly with neural recordings. For instance, future work can estimate Lyapunov spectra or other dynamical fingerprints from long, high‑resolution trajectories. We recognize that such an endeavor is technically demanding, given the need for stable, extended observations, and we will highlight this avenue for future work in the discussion if accepted.

---

> > ### Comment · Reviewer_ga7q · 2025-08-01
> >
> > I thank the authors for their response. All of my concerns have been addressed. I have no further questions and will raise my score to accept this paper.

---

### Official Review · Reviewer_zXGE · 2025-07-02

**Clarity:** 3
**Significance:** 3
**Originality:** 3
**Rating:** 4
**Confidence:** 4

**Summary:**

This paper focuses on chaos in finite-size RNNs with heavy tailed weight distributions. Authors first show that the transition to chaos in finite-size heavy tailed RNNs is sharp and depends on the network size. They relate the empirically observed extended critical regime to the largest Lyapunov exponent of the system. Finally, authors show that heavy-tailed RNNs in the critical regime present lower-dimensional chaotic dynamics the heavier the weight distribution tail is.

**Questions:**

- How do the results relate to task/information processing in recurrent networks? In particular, how doe s the extended critical regime and lower dimensionality interact with information processing at large, whether for reservoir computing or for a trained network? If the authors can provide a better link to task performance I would reconsider my evaluation score.
- Can the authors provide support for the claim regarding the tradeoff between dimensionality and robustness?
- In Figure 3c, the participation ratio increases with \alpha for strongly chaotic regime (large g). However for small g, near the transition (g<1) it appears the trend is reversed. What is the intuition behind this?
- Can the authors provide intuition for the definition of the Lyapunov dimension (Eq 4)? In their results it seems like the first term is dominant, is that right?

**Ethical Concerns:**

["NO or VERY MINOR ethics concerns only"]

**Final Justification:**

The authors have responded to my technical questions and they have motivated their work better in the context of ML, including a new benchmark on an XOR task, which is a fair compromise for my request to better understand how their heavy tailed weight initializations affect task performance. I have therefore raised my score by a point.

**Limitations:**

The authors have well identified the limitations of their work, in particular the open question of how the extended critical regime and lower dimensionality interact with information processing, be it in a reservoir computing framework or any type of trained RNN.

**Quality:**

3

**Strengths And Weaknesses:**

Strengths:
The submission is technically very solid and clearly written. Section 4.1 in particular is an elegant piece of work! While the transition to chaos in heavy tailed networks has been studied previously (Wardak and Gong, PRL 2022), the results on dimensionality via the PR make this work novel and interesting.

To the best of my knowledge, this sharpness-vs-size phenomenon for heavy-tailed RNNs has not been documented yet.

One additional contribution of the work (Line 95, point b) is to show the dependence of the transition point with N, which could be added to the list of contributions in line 63.

Weaknesses:
Overall the major weakness of the paper is the relevance for the NeurIPS community. The study appears to be directed primarily to a statistical physics audience, rather than to the machine learning community. No downstream information-processing or task-performance results are reported.

Other weaknesses include:
- More references to works in NeurIPS or other ML conferences would be appreciated.
- Line 64 could be misleading, as the authors reveal the nature of the transition, not the existence of the transition itself.
- In Figure 2a, it is unclear that the yellow and green dashed lines are overlapping. The authors could use transparencies to make this clear.
- The discussion, especially lines 345-362, is overstated and could be more precise.

---

> ### Author Rebuttal · Authors · 2025-07-29
>
> We deeply appreciate the reviewer’s detailed feedback and their high opinion on the elegance of Section 4.1 and the general clarity & solidity of our work. Reviewers zXGE & PCeb both point out that we present the first documentation of sharpness-vs-size phenomenon for heavy-tailed RNNs.
>
> The reviewer’s main concerns are regarding related extension, e.g., how the results tie to info processing, and suggests including more ML literature to reach a broader audience. Here we state the preliminary result on the robustness of heavy-tailed weights benefits reservoir computing performance in a temporal XOR task as a proof-of-concept, along with additional edits and intuition they kindly suggested. **Together with the discussion below, does this sufficiently address the reviewer’s concerns?**
>
> > Overall the major weakness of the paper is the relevance for the NeurIPS community…directed primarily to a statistical physics audience, rather than to the ML community. No downstream information-processing or task-performance results are reported.
>
> > More references to works in NeurIPS or other ML conferences would be appreciated.
>
> > How do the results relate to task/information processing in RNNs?...whether for reservoir computing or for a trained network? If the authors can provide a better link to task performance I would reconsider my evaluation score.
>
> We sincerely appreciate the constructive feedback aimed at broadening our reach and the opportunity to better situate our work in ML. We would like to respectfully clarify why we think the current work is directly relevant to ML, and how we will strengthen the relevance following the feedback:
>
> **(i) Downstream info-processing**
>
> While the core aim is to provide a principled & general characterization of the dynamical regimes induced by heavy-tailed connectivity (common in biological systems but underexplored in ML), we recognize the value of linking these results to info processing.
>
> If accepted, **we will include our new result on a delayed memory XOR task under a reservoir computing setup** (see task description in response to Reviewer V5Z3). Here, we observe that heavy-tailed networks exhibit a broader gain regime with strong performance compared to Gaussian networks. This aligns with our main results showing heavier-tailed networks exhibit a slower, more robust transition to chaos, which translates to task benefits here. This would provide a concrete proof-of-concept that the extended critical regime can enhance robustness and performance without fine-tuning, beneficial in ML.
>
> We hope the reviewer would find this addition a good middle ground between an extensive study of info processing (which we think is best-suited for future work) v.s. the current version of the paper, and would reconsider the evaluation score given the proposed change & below.
>
> **(ii) Current relevance to ML and NeurIPS**
>
> We respectfully argue that our study on the edge-of-chaos regime in untrained networks is highly relevant to ML, with precedents cited: prior work establishes the edge of chaos as a key initialization regime for network trainability and information propagation in both feedforward and recurrent networks (cited in lines 36-41).
>
> If accepted, we will integrate more references to further emphasize the ML connection, with the existing ones:
>
> * Schoenholz et al. (ICLR, 2017; **cited**) analyzed **untrained** random networks, showing that the trainability of deep networks depends on initializing near the edge of chaos; the farther from this critical regime, the shallower a network must be to remain trainable.
>
> * Bertschinger et al. (NeurIPS, 2004; **cited**) proposed that only near the edge of chaos can RNNs perform complex computations on time series.
>
> * Hayou et al. (ICML, 2019) extended Schoenholz et al. (2017) by tuning initialization and activation functions along the edge of chaos, improving training efficiency and performance.
>
> * Yang & Schoenholz (NeurIPS, 2017), with mean-field theory, show that residual connections enable deep networks to operate near the edge of chaos, thereby preserving input space geometry and improving stability across depth.
>
> * The topics are still highly relevant to ML, e.g., regularizing Lyapunov exponents can significantly enhance the effectiveness of RNN training (Engelken, NeurIPS, 2023); the expressivity and trainability of the Fourier Neural Operator peak at the edge of chaos, as shown via mean-field theory (Koshizukae et al., NeurIPS, 2024).
>
> We build directly upon these foundational insights. We show that heavy-tailed weight distributions naturally *broaden* the edge-of-chaos regime, an attribute desirable for training (above). While we draw upon tools from statistical physics, we believe the insights are relevant and our practice is fairly common in ML, e.g., mean-field analysis, energy landscapes, & scaling laws.
>
> We also emphasize the novelty of our contribution: theoretical understanding of heavy-tailed weights in finite-size networks remains limited due to analytical challenges. Our results provide a tractable & general framework for characterizing the dynamical behavior of such networks. We hope this foundational work will spark community interest in further investigations in ML.
>
> > Line 64 could be misleading, as the authors reveal the nature of the transition, not the existence of the transition itself.
>
> We respectfully clarify that, to our knowledge, the **existence** of the transition in finite-size networks is not a previously known result (lines 102-106) and **is first revealed in this work** (Section 4.1). The transition does not exist for the `tanh` activation function in the infinite-size limit under a mean-field theory (lines 103-104).
>
> > In Fig 3c, the participation ratio increases with \alpha for strongly chaotic regime (large g). However for small g, near the transition (g<1) it appears the trend is reversed. What is the intuition behind this?
>
> Intuitively, this is due to the early but slow transition of heavy-tailed networks. As shown in Fig. 2, heavier-tailed (smaller $\alpha$) networks undergo the transition to chaos at lower values of $g$, but the transition is slower. Thus, for small $g$, heavier-tailed networks have entered the weakly chaotic regime, whereas Gaussian networks remain in the quiescent regime. This explains why the attractor dimensionality can be higher for smaller $\alpha$ at low $g$: the heavier-tailed network has already developed some dynamical richness.
>
> > Can the authors provide support for the claim regarding the tradeoff between dimensionality and robustness?
>
> The tradeoff between dimensionality & robustness can be supported by interpreting Fig 2 (robustness) and Fig 3 (dimensionality) together, where only the tail-index $\alpha$ varies. The intention is to show that RNNs of smaller $\alpha$ (heavier tail) exhibit a slow transition to chaos (more robust to changes in $g$) but have a lower participation ratio & Lyapunov dimension (lower dimensionality).
>
> We interpret this as a tradeoff because of the opposite effects each is expected to have on information processing. While placing a system closer to the edge of chaos should allow it to integrate information along longer timescales (in RNNs) or across more layers (in feedforward networks), reducing the dimensionality of the attractor is expected to lower the effective dimensionality of the tasks for which these benefits can be observed. This is because information can only be effectively passed along the chaotic attractor, whereas any information encoded within directions associated with very negative Lyapunov exponents is effectively erased due to rapid contraction.
>
> > One additional contribution of the work (Line 95, point b) is to show the dependence of the transition point with N, which could be added to the list of contributions in line 63.
>
> Thanks! We will modify point b to “We reveal that…, *with the locations of the transition depending on network size N.*”
>
> > Can the authors provide intuition for the definition of the Lyapunov dimension (Eq 4)? In their results it seems like the first term is dominant, is that right?
>
> Yes, the 1st term is dominant. The Lyapunov dimension (LD) is related to the Kaplan-Yorke conjecture that states, for ‘typical attractors’, Lyapunov exponents (LEs) can be used to estimate the fractional (information) dimension. Intuitively, it gives the number of dimensions needed to describe dynamics on the attractor. The positive (negative) LEs correspond to directions in which a small cloud of points is expanded (contracted). In the definition of the LD, $k$ denotes the largest integer such that the sum of the top LEs is still positive. Thus, if we follow an initial $k+1$-dimensional cloud of points generated around a point on the attractor, we will notice that the overall volume of the cloud tends to contract on the attractor (since the sum of Lyapunov exponents is negative). In contrast, if we instead follow a $k$-dimensional cloud of points associated with the top $k$ LEs, the sum is (usually) still positive, so the cloud tends to expand. The expansion and contraction are balanced when, on average, the number of dimensions is equal to the LD.
>
> The 2nd term estimates the fractional part of the information dimensions and lies between $0$ and $1$, so the first term indeed dominates in our analysis of high-dimensional chaotic activity.
>
> > In Fig 2a, it is unclear that the yellow and green dashed lines are overlapping. The authors could use transparencies to make this clear.
>
> Thanks. We will tune the transparencies to improve visualization.
>
> > The discussion, especially lines 345-362, is overstated and could be more precise.
>
> Thanks. We will word the discussion more carefully, e.g.:
> * Instead of saying “supports the idea” (line 351), reword to “provides one piece of evidence”.
> * Instead of saying “are directly testable” (line 360), edit to “offer inspirations for future works”.

---

> > ### Comment · Reviewer_zXGE · 2025-08-02
> >
> > Thanks to the authors for their technical clarifications and for the extra effort to train the XOR task and make links to the broader ML community. I have no further questions and will raise my score.

---

> ### Comment · Area_Chair_iJhY · 2025-08-01
> **Relevance**
>
> In my opinion, the paper is perfectly suited for the subject matter (theoretical neuroscience) which has historically been an element of NeurIPS. Can you adjust your score based on this, and in light of the authors' rebuttal?

---

### Official Review · Reviewer_PCeb · 2025-07-03

**Clarity:** 4
**Significance:** 4
**Originality:** 4
**Rating:** 5
**Confidence:** 4

**Summary:**

The paper investigates recurrent neural networks (RNNs) whose synaptic weights are drawn from Lévy $\alpha$-stable (heavy-tailed) distributions.
The authors derive a finite-size theory for the critical gain $g^\ast$ at which an autonomous network leaves the quiescent fixed point and becomes chaotic.
They also confirm the prediction in simulations and demonstrate that heavier tails ($\alpha<2$) broaden the “edge-of-chaos’’ regime, producing a slower, more robust transition and reveal a trade-off heavier tails compress activity onto lower-dimensional attractors, quantified by smaller Lyapunov and participation-ratio dimensions.

**Questions:**

1. Can the authors discuss more about how robust the critical-gain prediction is when the annealed assumption is removed? Can the authors bound the variance of $g^\ast$ across quenched realizations analytically?
2. Do the authors believe that robustness stems from global heavy-tail statistics or from extreme outliers? An $\alpha$-stable law has diverging variance, so a handful of ''mega-synapses’’ could be driving the dynamics. Do the authors have a sense of which scenario better accounts for the observed phenomena? One way to probe this would be by removing the top few weights (or clipping them) and seeing whether that shifts the critical gain and attractor dimension.
3. How easily can this analysis be extended to a setting with mixed $\alpha$ weights (i.e., a subset of weights with $\alpha_1$, and another with $\alpha_2$, etc.)? A full investigation would likely be best for a follow-up work, but it seems like an interesting extension that may be worth commenting on to improve the model's biological plausibility.

**Ethical Concerns:**

["NO or VERY MINOR ethics concerns only"]

**Limitations:**

Yes

**Quality:**

4

**Strengths And Weaknesses:**

**Strengths**:
This is a well-written paper that addresses an important gap in the literature on heavy-tailed RNNs, which are of significant interest for several reasons, chief among them being modeling biological and neural connectivities.
The results are clearly presented and substantiated, and the results of the analysis are carefully interpreted to reveal insights about how heavy tails impact finite-size networks (e.g., clear quiescent-to-chaotic transition).
It is a compelling instance showing how mean-field analyses falls short of characterizing an interesting phenomenon.

To my knowledge, this is indeed the first finite-size theory for heavy-tailed RNNs, and it yields interesting results.
The authors provide solid empirical support with extensive sweeps over $g$, $\alpha$, and $N$ validate theory and expose non-trivial phenomena (e.g.\ non-monotonic behaviour for $\alpha<1$).

**Weaknesses**:
The analysis relies on an annealed approximation to the true, quenched, setting.
There is a significant gap between the behavior of the annealed and quenched settings, as shown in Figure 1, especially for smaller $N$.

---

> ### Author Rebuttal · Authors · 2025-07-29
>
> We deeply appreciate the reviewer’s thoughtful feedback and their high opinion on the significance, solidity, and clarity of our work, especially pointing out that, to the best of their knowledge, our work presents the first finite-size theory for heavy-tailed RNNs (along with Reviewer PCeb).
>
> The reviewer mainly has questions regarding the interpretation of what drives robustness, the setting and assumptions of our theoretical analysis, and an interesting extension of the work to models of mixed heavy-tailed weight distributions. Here, we address each, with additional experiments on probing whether outlier weights are critical to the observed phenomena. **Together with the discussion below, does this sufficiently address the reviewer’s questions?**
>
> > Do the authors believe that robustness stems from global heavy-tail statistics or from extreme outliers? An $\alpha$-stable law has diverging variance, so a handful of ''mega-synapses’’ could be driving the dynamics. Do the authors have a sense of which scenario better accounts for the observed phenomena? One way to probe this would be by removing the top few weights (or clipping them) and seeing whether that shifts the critical gain and attractor dimension.
>
> We thank the reviewer for their spot-on insight! To probe this, we ran the simulations in Figs. 2 and 3, with four settings: we removed (i) top 1%, (ii) top 3%, (iii) bottom 90%, (iv) bottom 95% of recurrent weights by the absolute magnitude.
>
> We found the results largely indicate it is a handful of “mega-synapses” driving the dynamics, implying **the robustness indeed largely stems from *extreme outliers.*** Specifically, we found the slow transition is no longer present in cases (i) and (ii), and the transition points are shifted to larger $g^\*$ (the more top outliers pruned, the larger $g^\*$ becomes). The slow transition persists in cases (iii) and (iv) when the outliers remain but the rest of the weights are pruned.
>
> Similarly, removing the bottom percentage of weights has very little effect on the general trend of attractor dimensionality as presented in the main paper. However, when top outliers are removed, the changes are more nuanced: we observe that the attractor dimensionality for heavy-tailed weights increases in the chaotic regime, while the transition to chaos is pushed to a larger $g^\*$ when more top outlier weights are pruned as mentioned above. The general ranking of dimensionality by $\alpha$ is largely consistent with the main paper for both participation ratio and Lyapunov dimension, though the max dimensionality of $\alpha=1.5$ is comparable to that of $\alpha=2$ over a range of $g$ when top outliers are pruned.
>
> Given no external links can be included per the submission rule, we state the results here and will include the figures in the manuscript if accepted.
>
>
> > The analysis relies on an annealed approximation to the true, quenched, setting. There is a significant gap between the behavior of the annealed and quenched settings, as shown in Figure 1, especially for smaller N.
>
> > Can the authors discuss more about how robust the critical-gain prediction is when the annealed assumption is removed? Can the authors bound the variance of across quenched realizations analytically?
>
> Thanks for the insights! Inspired by the reviewer’s comment, we ran additional simulations to check how realization-to-realization fluctuations of the transition point change with the size of the network $N$. For simplicity, we focus on $\alpha=1.0$. In the manuscript, we showed that the annealed theory predicts scaling $g^\* \sim 1/\ln(N)$. In the case of quenched weight disorder, when averaged over realizations of the weight matrix, $\langle g^\* \rangle$ also seems to behave like $\sim1/\ln(N)$ and it stays close to the annealed theoretical prediction for all $N$, with a small positive bias that slowly decreases with $N$. The standard deviation also decreases with increasing $N$.
>
> In order to characterize the relative strength of the fluctuations, we additionally plotted the coefficient of variation as a function of $N$. It showed a significant decreasing trend, indicating that in the studied range ($N$ between $32$ and $3000$), the standard deviation decreases faster than the mean. Similarly, the relative mean squared error (i.e., the MSE between annealed and quenched transition $g$ divided by the squared mean annealed transition $g$) decreased with $N$. These results suggest that the dynamics becomes ‘more self-averaging’ and the annealed approximation becomes better with $N$. However, these empirical observations do not give us certainty that the same qualitative trends continue for even larger networks. Unfortunately, the mathematical analysis of the finite-size effects in the quenched networks is beyond the tools used in this work, and we were not able to obtain mathematical results bounding the variance across quenched realizations.
>
> To summarize, as expected, the annealed theory is an approximation of the behavior of the quenched system (it is exact for feedforward networks, but not exact for RNNs). But this approximation predicts quite well the average behavior of even small quenched systems and does seem to get better as $N$ increases.
>
> We plan to include the results of this numerical experiment in the Appendix of the revised version of the manuscript.
>
> > How easily can this analysis be extended to a setting with mixed weights (i.e., a subset of weights with $\alpha_1$, and another with $\alpha_2$, etc.)? A full investigation would likely be best for a follow-up work, but it seems like an interesting extension that may be worth commenting on to improve the model's biological plausibility.
>
> We thank the reviewer for this very insightful and constructive suggestion. We agree that a mixture setting is an interesting avenue and could enhance biological plausibility, and that a full investigation would be best suited for future work. We believe our framework can indeed be extended to this setting, though there are multiple possibilities to consider: e.g., the model can
>
> (i) be homogeneous, where each neuron’s weight is chosen randomly from one of $\alpha_1$ or $\alpha_2$ distributions, or
>
> (ii) have two (or more) interacting populations, where each population has a distinct $\alpha$ and projects differently within or across groups—this could biologically correspond to different cell types with distinct synaptic statistics and connectivity patterns.
>
> We will include this point in our Discussion to highlight the potential for future exploration.

---

> > ### Comment · Reviewer_PCeb · 2025-08-05
> >
> > Dear Authors,
> >
> > I thank you for your very detailed responses to each of my questions, in particular on the validity of the approximation. I strongly support acceptance and maintain my high score.

---

### Decision · Program_Chairs · 2025-09-17

**Decision:**

Accept (poster)

**Comment:**

This paper discusses dynamics of RNNs. Given that I have only worked on this area briefly and a long time ago, I do not feel competent to comment. However, since they are all positive, I recommend acceptance.